# Predicted antiviral potential of phytochemicals prolific in *Cleistanthus bracteosus* Jabl. and essential oils of *Artemisia scoparia* and *Thuja orientalis* against Nipah virus and Human metapneumovirus: An AI-driven *in-silico* study

Mackingsley Kushan Dassanayake[1]*, Teng-Jin Khoo[1], Chien Hwa Chong[2]*, Mohammed Tahir Ansari[1], Patrick Di Martino[6], Fernando Berton Zanchi[8], Adam Figiel[3], Antoni Szumny[4], Omar Ashraf Elfar[9], Christophe Wiart[5], Rachael Symonds[7]

1 School of Pharmacy, Faculty of Science and Engineering, University of Nottingham Malaysia, Jalan Broga, Semenyih, Malaysia, 2 Department of Chemical and Environmental Engineering, Faculty of Science and Engineering, University of Nottingham Malaysia, Jalan Broga, Semenyih, Malaysia, 3 Institute of Agricultural Engineering, Wrocław University of Environmental and Life Sciences, Wrocław, Poland, 4 Department of Chemistry, Wrocław University of Environmental and Life Sciences, Norwida, Wrocław, Poland, 5 Institute of Tropical Biology and Conservation, University Malaysia Sabah, Kota Kinabalu, Sabah, Malaysia, 6 BCMI Research Group, ERRMECe Laboratory, Cergy Paris University, Cergy-Pontoise, France, 7 School of Biological and Environmental Sciences, Faculty of Science, Liverpool John Moores University, James Parsons Building, Liverpool, United Kingdom, 8 Laboratório de Bioinformática e Química Medicinal, Fundação Oswaldo Cruz Rondônia (LABIOQUIM-Fiocruz-RO), Porto Velho, Rondônia, Brazil, 9 Department of Pharmaceutics and Industrial Pharmacy, Faculty of Pharmacy, Cairo University, Kasr El-Aini, Cairo, Egypt

* ChienHwa.Chong@nottingham.edu.my (CHC); rapter100000@yahoo.com (MKD)

## Abstract

The recent Nipah virus (NiV) epidemic and human metapneumovirus (hMPV) outbreak have had a significant impact on human health and society worldwide. The attachment glycoprotein (G) and fusion glycoprotein (F0) of NiV and hMPV are essential for pathogenesis and are potentially pronounced targets for antiviral treatment. In the present study, we utilised computational methods to analyse the predictive antiviral potential of phytochemicals present in *Cleistanthus bracteosus* and in the essential oils of *Artemisia scoparia* and *Thuja orientalis* against NiV and hMPV. Molecular docking and dynamics simulations were the primary tools for assessing the binding interactions of compounds detected by GC-MS. Three out of four compounds tested (digoxigenin, cedrene and cedrol) exhibited remarkable binding affinities between −7.7 kcal/mol and −6.2 kcal/mol for NiV fusion glycoprotein (F0), and between −8.3 kcal/mol and −7.1 kcal/mol for NiV attachment glycoprotein (G). Similarly for hMPV fusion glycoprotein (F0), the aforesaid compounds showed binding affinities between −8.1 kcal/mol and −6.4 kcal/mol. Moreover, MD simulations illustrated phytochemical interacting amino acid residues associated with each receptor of NiV and hMPV. These phytochemical compounds were further evaluated using ADMET platforms. In conclusion, the present *in silico* work predicts for the first

**Data availability statement:** All relevant data are within the manuscript and its Supporting Information files.

**Funding:** The author(s) received no specific funding for this work.

**Competing interests:** The authors have declared that no competing interests exist.

**Abbreviations:** *C. bracteosus*: *Cleistanthus bracteosus*; NiV: Nipah virus; hMPV: Human metapneumovirus; SARS-CoV-2: Severe acute respiratory syndrome coronavirus 2: HSV-1: Herpes simplex virus 1; 2VWD: NiV attachment glycoprotein (G); 5EVM: NiV fusion glycoprotein (F0); 5WB0: hMPV fusion glycoprotein (F0); EO: Essential oils; *A. scoparia*: *Artemisia scoparia*; *T. orientalis*: *Thuja orientalis*; GC-MS: Gas chromatography-mass spectroscopy; MD: Molecular dynamics; μL: Microlitres; mL: Millilitres; eV: Electron volt; M/Z: Mass-to-charge ratio; mL/min: Millilitre per minute; mg/kg bw/day: Milligram per kilogram body weight per day; ug/L: Microgram per litre; mol/L: Mol per litre; L/kg: Litre per kilogram; kJ/mol/nm: Kilojoule per mol per nanometre; RRT: Relative retention time; rpm: Revolutions per minute; TLC: Thin layer chromatography; kcal/mol: Kilocalorie per mole; ns: nanoseconds; nm: nanometres; atm: Atmosphere; WHO: The World Health Organization; PDB: Protein Data Bank; SDF: Structured Data File; RBD: Receptor binding domain; RMSD: Root mean square distance; RMSF: Root-mean-square fluctuation; RNA: Ribonucleic acid; ADME: Absorption, distribution, metabolism and excretion; CNS: central nervous system; hERG: Human ether-a-go-go gene; NVT: Constant number of atoms (N), constant volume (V), and constant temperature (T); NPT: Constant number of atoms (N), constant pressure (P), and constant temperature (T); AMBER: Assisted Model Building and Energy Refinement; GROMACS: GROningen MAchine for Chemical Simulations; 2VWD: NiV attachment glycoprotein (G); 5EVM: NiV fusion glycoprotein (F0); 5WB0: hMPV fusion glycoprotein (F0); HIV: Human immunodeficiency virus; H1N1, H3N2, H5N1: Influenza A virus subtypes; Rg: Radius of gyration; SASA: Solvent-accessible surface area.

time the predicted potential of using major compounds present *C. bracteosus, A. scoparia* and *T. orientalis* as a novel anti-viral therapeutic strategy to control the entry and pathogenesis of NiV and hMPV. Despite few RMSD fluctuations in protein-ligand complexes stemming from structural alterations in the beta-turn-beta and helix-coil-helix, the simulations remain mostly stable from 50 ns till 100 ns.

## Introduction

The emergence and development of disease-causing viruses posed a phenomenal threat to human health and became an enormous challenge to modern medicine and the global economy. These mainly zoonotic viruses can be the cause of epidemics or even pandemics. [1–3]. The Nipah virus (NiV) is an example of such outbreak causing pathogenic agent, which has been categorised by WHO as a potential cause of epidemics in the future [4,5].

The NiV is a zoonotic henipavirus, which was first discovered more than two decades ago after an outbreak among pig farmers in Malaysia [6]. Within months, it spread to Singapore through infected pigs. The epidemic resulted in almost 300 cases and more than 100 deaths. Since then, no other outbreaks of NiV have been reported in Malaysia. But in 2001, the virus appeared in Bangladesh and India, where outbreaks continued intermittently. The virus can cause fever, vomiting, breathing problems and inflammation in the brain. It is mostly carried by fruit bats, but it can also infect domestic animals such as pigs, as well as humans. It is spread by contact with bodily fluids of infected animals or humans. There are no approved vaccines or treatments, but researchers are investigating candidates. Outbreaks occur almost annually in Bangladesh, and studies have linked infections to drinking fermented date juice contaminated with urine from bats [7]. Two major clades of NiV are responsible for most of the reported outbreaks. The NiV isolates from India and Bangladesh belong to the NiV-BD clade, whereas the NiV isolates from Malaysia and Cambodia are clustered in NiV-MY clade. While the Malaysian strain spread from animals to humans, there were minimal human-to-human transmissions [8–11]. The NiV genome consists of a single-stranded, negative-sense RNA that encodes six structural proteins, phosphoprotein (P), nucleoprotein (N), fusion protein (F), matrix protein (M), attachment glycoprotein (G), and RNA polymerase protein (L) or the large protein [12]. The primary receptors for the Nipah virus are the cell-surface proteins ephrin-B2 and ephrin-B3 found on host cells, by which cellular attachment is assisted via glycoprotein (G) and fusion protein (F) expressed on the virion. These transmembrane proteins play roles in cell signalling, the development of blood vessels and neurogenesis [13]. Since there is no specific treatment for Nipah virus infection yet, it is relevant to explore the therapeutic potential of plants used in traditional medicine.

Human metapneumovirus (hMPV) is a relatively recent addition to the spectrum of respiratory pathogens, first identified in the Netherlands in 2001 [14]. This virus belongs to the subfamily *Pneumovirinae* and is classified under the genus Metapneumovirus [15]. As a non-segmented, negative-sense RNA virus from the

*Pneumoviridae* family, its genome spans approximately 13 kilobases and comprises eight genes (N, P, M, F, M2, SH, G, and L). These genes encode nine proteins, including three crucial surface glycoproteins: fusion (F), small hydrophobic (SH), and glycosylated (G) proteins that facilitate cellular attachment and entry [16]. The hMPV infects and proliferates extensively within airway epithelial cells associated with the respiratory system, primarily via RGD-binding integrins like αVβ1 and α5β1 receptors found on host cells [17].

Phylogenetic studies have established a strong evolutionary link between hMPV and avian pneumoviruses. Research suggests that hMPV likely originated from an avian reservoir, with genetic evidence pointing to a common ancestor shared with avian metapneumovirus subtype C (AMPV-C) [18]. The virus affects individuals of all age groups, from infants as young as two months to elderly individuals up to 87 years of age [19].

Over the years, significant advancements have been made in understanding the global burden of hMPV. Hospitalisation rates associated with hMPV infections in children under five years of age are estimated at approximately 1 per 1,000 annually [20]. Notably, it took over two decades after the discovery of AMPV for hMPV to be identified [21].

Recent studies have refined the classification of hMPV into five genotypes: A1, A2a, A2b, B1, and B2, with genetic diversity primarily observed in the G gene, the most variable region among hMPV strains. Novel genetic variants featuring 111- or 180-nucleotide duplications in the G gene have led to the emergence of new clades, designated A2b1 and A2b2 (or A2c) [22]. However, a recent phylogenetic analysis of whole-genome hMPV sequences suggests that A2b2 and A2c are likely the same subtype, highlighting the necessity for a standardised nomenclature reassessment by the International Committee on Taxonomy of Viruses (ICTV) [23].

Despite advancements in characterising the virus, there is currently no approved antiviral treatment for hMPV infections. Symptomatic relief is typically achieved with over-the-counter medications, as most individuals recover within a few days [24]. Continued research focusing on the molecular and phenotypic distinctions between AMPV-C and hMPV is essential for the development of a targeted vaccine. Throughout the years, gathering traditional knowledge from local and indigenous communities, alongside harnessing the power of ethnopharmacological plants to extract bioactive molecules and phytochemicals, presents a compelling yet challenging approach to diagnosing a range of ailments [25]. Many factors, including the choice of solvents—whether polar or non-polar—and the selection of specific plant parts or tissues, play a pivotal role in the effective extraction of these vital bioactive constituents [26]. To harness these bioactive molecules against viral infections, a holistic strategy for their isolation and characterisation, coupled with virus replication inhibitory experiments in animal cell systems, is not just beneficial but essential [27]. The enduring ambition of establishing high-throughput screening assays serves as a pathway to swiftly and accurately identify bioactive molecules and phytochemicals from vast chemical libraries. Ultimately, rigorous *in vivo* experiments and subsequent clinical studies are crucial for assessing the antiviral potential and potential complications—such as reactogenicity or toxicity—of the purified bioactive compounds [28].

The advent of expansive chemical repositories and combinatorial chemical spaces, paired with the innovative power of high-throughput docking and generative AI, has dramatically enriched the landscape of small molecule diversity for drug discovery. The meticulous curation of these vast chemical databases, alongside breakthroughs in algorithms and computational hardware, has significantly amplified the scale and potential of in silico drug discovery campaigns. In the pursuit of finding compounds for experimental validation, it becomes vital to filter these molecules based on desirable drug-like properties, such as Absorption, Distribution, Metabolism, Excretion, and Toxicity (collectively known as ADMET). The ADMET platform stands at the forefront of this endeavour, expertly crafted to predict these essential properties in molecules. Harnessing the power of advanced machine learning models, including cutting-edge graph neural networks like the Chemprop-RDKit model, it is trained on extensive datasets to efficiently screen large chemical libraries. Renowned for its remarkable speed and precision, the ADMET platform is not just a tool; it is an invaluable ally in the quest for groundbreaking drug discovery and development [29].

*Cleistanthus* spp., *A. scoparia* and *T. orientalis* have traditionally been used to treat a variety of illnesses and conditions in a wide range of Middle Eastern and East Asian nations, which have medicinal applications to treat minor headaches,

insomnia, fever, palpitation and promotion of hair growth and acute gastrointestinal disorders in folklore medicine. These medicinal properties have been validated experimentally by several investigations [30–32]. It has been reported that *A. scoparia* and *T. orientalis* were used to treat influenza-like respiratory infections in folklore medicine. Meanwhile, phytoconstituents present in *Cleistanthus* spp., has been disclosed to have antiviral potential against respiratory viruses [33–35].

Previous studies have proved the antibacterial potential of extracts derived from *Cleistanthus* spp., *A. scoparia* and *T. orientalis* against multi-drug-resistant (MDR) bacteria, which were potentially suitable drug candidates against several pathogens [36–38]. Preliminary phytochemical analysis has shown the existence of tannins, terpenoids, flavonoids, saponins, glycosides, steroids, phlobatannins, and alkaloids that are responsible for anti-HIV-1 of *Cleistanthus* spp. Meanwhile, the inhibitory effect against influenza A by *T. orientalis* was presumed on certain compounds such as amentoflavone, myricetin, quercetin, and quercitrin. Moreover, scopoletin, arteannuin B, and artemisinic acid of *Artemisia* spp. have shown to suppress the replication of SARS-CoV-2 [39–41].

The development of information technology permeates many aspects of drug development today. Such methods include *in silico* analysis for hit detection and lead optimization techniques that promote low-cost and safe screening of potential agents for rational drug development. *In silico* analysis can be classified into structure-based and ligand-based methods [42]. Molecular docking has been the most commonly used technique for *in silico* analysis of molecular interactions since the early 1980s. Computer programs based on various algorithms have been developed to perform molecular docking studies, as this *in silico* technique has become an increasingly important and critical tool in drug discovery, in particular, for the identification of potential anti-virals [43]. This study was conducted to identify new broad-spectrum naturally occurring plant-based antiviral agents that target the attachment of the virus particle and its entry into the target cell. The selection of these plants is justified by the presence of phytochemicals, which are generally known to provide health benefits. These include bioactive nutrients that have demonstrated the ability to lower the risk of significant chronic diseases in humans. Research from preclinical, clinical, and epidemiological studies has indicated that phytochemicals may effectively treat various ailments, thanks to their antioxidant and anti-inflammatory properties [39–41]. Here, we describe the binding potential of bioactive secondary metabolites naturally produced by plants known as *Cleistanthus bracteosus* Jabl., *Artemisia scoparia* and *Thuja orientalis* to attachment glycoprotein (G) and fusion glycoprotein (F0) of Nipah virus and with the fusion glycoprotein (F0) of human metapneumovirus using molecular docking computer programs in a virtual environment as a pioneering study. Additionally, this study motivates the exploitation of research on the discovery of novel compounds and *in vitro*, *in vivo* or *ex vivo* probing of the selected plants for studies relevant to anti-NiV and anti-hMPV activities.

## Materials and methods

### Ethics approval and consent to participate

The study did not require the consent of the participants.
Ethics declaration: not applicable.

### Collection and authentication of plant material

Leaves, bark, stem and wood of *Cleistanthus bracteosus* were collected from Manong primary rain forest in Perak, Peninsular Malaysia (GPS coordinates: 4°72″N 100°81″E) on 3rd February 2017. A sample of the whole plant was sent to the Forest Research Institute of Malaysia (FRIM) located in Kepong, Selangor for authentication. A voucher sample of the whole authenticated plant of *C. bracteosus* (NB125) was deposited under the Forest Research Institute of Malaysia (FRIM).

Whereas samples of *Artemisia scoparia* and *Thuja orientalis* were purchased from the local florist, Fong Seng Landscape Trading (https://g.co/kgs/jSx5qE) in Malaysia on 28th May 2023.

**Preparation and extraction of *C. bracteosus*.** The rotor beater mill (Retsch, Düsseldorf, Germany) and the ultra-centrifugal mill (Retsch, Düsseldorf, Germany) were used to fragment clean, air-dried bark samples of *C. bracteosus*. By using a porcelain mortar and pestle, the broken pieces were ground into a coarse powder. One litre (1 L) of chloroform EMSURE grade (Sigma-Aldrich, St. Louis, USA) was used to dissolve seventy grams (70 g) of the powdered bark. The combination, which contained plant material powder in a 1:14 (w/v) solvent ratio, was macerated for seven days at 40°C in a water bath before spending an additional two days in an orbital shaker. Sonication at 40°C for 30 minutes helped the maceration even further. Whatman grade 1 filter paper (GE Healthcare, Chicago, USA) was used to filter the macerated solution, and rotary evaporation under reduced pressure at 40°C was used to evaporate the solvent content of the macerated mixture. The chloroform crude bark extract of *C. bracteosus* was subjected to TLC fractionation, and the most bioactive fraction was centrifuged with dichloromethane HPLC grade (Merck, New Jersey, United States) at 9000 rpm for 10 min. Subsequently, the supernatant was filtered using a 0.22 μm pore PTFE syringe filter and taken for GC-MS analysis.

### Preparation and extraction of essential oils from *A. scoparia* and *T. orientalis*

Recently harvested leaves were washed and broken apart. The essential oils were extracted using the steam distillation method. The distillation flask, which had a 500 mL capacity, was filled with 150 g of mashed plant material. A glass tube and condenser connected the flask to the steam generator. Where, the flask holding plant material was filled with distilled water to approximately half of its capacity, and it was kept warm on a heating mantle between 90°C and 100°C. During the distillation process, the mixture that contained water and essential oil that evaporated from the plant material was collected in a different flask. Using a separatory funnel, the liquid mixture containing the essential oil was extracted from the aqueous phase following the steam distillation process, using diethyl ether EMSURE grade (Sigma-Aldrich, St. Louis, USA). Later, rotary evaporation under reduced pressure was used to remove diethyl ether and obtain pure essential oil. The EOs were mixed with dichloromethane HPLC grade (Merck, New Jersey, United States) and passed through a PTFE syringe filter of 0.22 μm pore size and subjected to GC-MS analysis.

### Determination of the most abundant compounds present in the selected plants using gas chromatography–mass spectrometry (GC-MS)

*C. bracteosus* chloroform fractionated bark extract, and the essential oils of *A. scoparia* and *T. orientalis* were separately dissolved in 100% dichloromethane HPLC grade (Merck, New Jersey, United States), and GC-MS was carried out on a PerkinElmer Clarus 680, 5Q85 mass spectrometer (Massachusetts, USA) using the Elite 5MS column (30 m × 0.25 mm × 0.25 μm film). An electron ionisation system with an ionisation energy of 70 eV was utilised for GC-MS detection. Scan range 35−600 m/z in mode of 0.5 scans s$^{-1}$. Helium was used as the carrier gas with a flow rate of 1 mL/min. Injector and oven temperatures were set at 250°C. The oven temperature was the same as with the GC analysis. The samples of 1 μL were injected in the 50:1 split mode.

The identification of compounds was performed by comparing their RRT relative retention time, peak area % and mass spectra-to-charge ratio with those of reference samples, literature/data and computerised MS–data bank NIST (National Institute of Standards and Technology) 2.0 (2005) database. The peak area method was followed for the quantitative assessment of different constituents, and the percentage was calculated relatively.

### Ligands and receptors executed for in-silico analysis

**Ligands.** A total of five (5) ligands, which include phytochemical compounds, cedrene (CID: 521207) from *A. scoparia,* cedrol (CID 65575) from *T. orientalis* and digoxigenin (CID 15478) and lauric acid (CID 3893) *from C. bracteosus*, and chloroquine (CID 2719) as the control, were selected for the molecular docking. The rationality for selecting the

aforementioned phytochemical compounds includes their high abundance in the selected plant extracts and their prominence of inducing a variety of broad-spectrum antimicrobial activities. The three-dimensional (3D) structures of the selected phytochemical compounds were downloaded from the National Library of Medicine (NLM) PubChem PDB database (https://pubchem.ncbi.nlm.nih.gov) accessed on 10 October 2024 in SDF format.

**Receptors.** The viral target receptors were NiV attachment glycoprotein (G) (PDB ID: 2VWD), NiV fusion glycoprotein (F0) (PDB ID: 5EVM) and hMPV fusion glycoprotein (F0) (PDB ID: 5WB0). The three-dimensional (3D) structure of the selected viral protein was downloaded from the Research Collaboratory for Structural Bioinformatics Protein (RCSB) PDB database (https://www.rcsb.org/) accessed on 10 October 2024 in PDB format.

**Modelling and preparation of selected macromolecules.** The PDB structures of NiV attachment glycoprotein (G), NiV fusion glycoprotein (F0) and hMPV fusion glycoprotein (F0) and SDFs of phytochemical compounds digoxigenin, lauric acid, cedrene, cedrol and the control reference chloroquine were modified using AutoDockTools (version 1.5.7, La Jolla, CA, USA), where water molecules were removed and hydrogen bonds and Kollman charges were added to their 3D structures. The modified SDF and PDB files were converted to Protein Data Bank, Partial Charge (Q), and Atom Type (T) (PDBQT) format before being analysed to optimise docking efficiency. The retrieved protein structures were validated using a Ramachandran plot generated by PROCHECK (version 3.5.4) to determine the permitted regions for torsion angles ψ against φ of amino acid residues (S1 Fig).

**Determining the physicochemical properties of attachment glycoprotein (G) and fusion glycoprotein (F0).** The physicochemical properties of the selected viral proteins, which were molecular weight, amount of negative and positive residues, isoelectric point, instability index, extinction coefficient, aliphatic index and grand average of hydropathicity (GRAVY) were determined via ProtParam, which was a tool utilised by ExPASy [44].

**Molecular docking analysis.** The molecular docking was performed by the application of AutoDock Vina (version 1.1.2, La Jolla, CA, USA) software. The blind docking was performed at a default grid box dimension of 40 Å × 40 Å × 40 Å and the energy range was set at 4 and exhaustiveness was set at 8 [45].

**Determination of the root mean square distance for binding efficiency.** AutoDock Vina software was used to calculate and estimate the docking energies. The Autodock Vina log file of the docked molecule indicating the binding affinity measured in kcal/mol is allocated to a root mean square distance (RMSD) value of zero was selected as the best result to confirm the most accurate output. The scoring function for AutoDock Vina was based on 6–12 van der Waals interactions and Coulomb energies, combined with Broyden-Fletcher-Goldfarb-Shanno algorithm [45].

**Simulation of molecular ligand-receptor interactions.** Simulation and visualisation of molecular interactions between the selected viral protein and phytochemical compounds indicating the active binding sites of ligand-receptor complexes and their amino acid sequences were performed using PyMOL (version 2.5.2) molecular visualisation system [46].

**Molecular dynamics simulation.** MD simulations were applied for validation of results obtained via molecular docking and offer a consistent assessment of the potential stability of a protein–ligand complex [47]. In this study, MD simulations were performed for NiV attachment glycoprotein (G), NiV fusion glycoprotein (F0) and hMPV fusion glycoprotein (F0) complexed with the selected phytochemicals presenting binding affinities ≤−6 kcal/mol that surpassed the cut-off value during molecular docking studies, which invigorated their docking compatibility. Molecular dynamics (MD) simulations were performed using the GROningen MAchine for Chemical Simulations (GROMACS, version 2024.1) [48] with interface Visual Dynamics [49] for generating scripts and Assisted Model Building and Energy Refinement (AMBER) 99 force field [50]. The partial charges and ligand topologies were obtained by Acpype [51] using the ANTECHAMBER [52] module. Electrostatic interactions were treated using the particle mesh Ewald (PME) algorithm with a cut-off of 12 Å. Each system was simulated under periodic boundary conditions in a cubic box, whose dimensions were automatically defined, considering 5 Å from the outermost protein atoms in all Cartesian directions. The simulation box was filled with TIP3P water molecules [53]. Subsequently, a two-step energy minimisation procedure was performed (2000 steps of

steepest descent and 2000 steps of conjugate gradient or until the system reaches a resistance force lower than 1000 kJ/mol/nm. Next, initial atomic velocities were assigned using the Maxwell-Boltzmann distribution corresponding to a temperature of 300 Kelvin (K) using the Langevin thermostat and Berendsen barostat systems. All systems were subsequently equilibrated during two successive NVT and NPT equilibration simulations with 200 ps for each. After this period, all the systems were simulated with no restraints at 300 K in the Gibbs ensemble with a 1 atm pressure using isotropic coupling. Poses were performed with a collection rate of 2 x 10$^{-15}$ (2 fs). All chemical bonds containing hydrogen atoms were restricted using the SHAKE algorithm [54], and the time step was set to 2 fs. Finally, we simulated MD runs of 100 ns for each complex.

Simulation trajectories were analysed with GROMACS package tools [48]. The root mean square deviation (RMSD) and the root mean square fluctuation (RMSF) were calculated separately for each system fitting their heavy atoms, taking the initial structure of the production dynamics as a reference. Hydrogen bond analysis and the radius of gyration (Rg) were examined using the output trajectory files obtained from the simulation. Additionally, calculating the solvent-accessible surface area (SASA) facilitated the analysis of the MD simulation results and assisted in identifying significant motions according to their amplitude.

**Calculation of binding free energy.** By employing GROMACS trajectories, the Molecular Mechanics/Poisson-Boltzmann Surface Area (MMPBSA) method was applied to determine the binding free energy of the complex. MM-PBSA was assessed according to the standard formula presented below:

$$\Delta G_{binding} = G_{complex} - (G_{protein\ receptor} - G_{ligand})$$

Where $G_{complex}$ was the total free energy of the ligand-receptor complex, $G_{protein\ receptor}$, and $G_{ligand}$ indicates the total free energy of the target and the drug individually [55,56].

**ADME, toxicity and drug likeness studies.** By the application of PkCSM pharmacokinetics AI (artificial intelligence) tool, ADME + T properties (Absorption, Distribution, Metabolism + Toxicity) were analysed and the toxicity of the leading compounds with binding affinities ≤ −6 kcal/mol (digoxigenin, cedrene and cedrol) and the comparative reference drug (chloroquine) were determined [57–59].

The drug likeness was evaluated for 90% of orally active drugs that have achieved phase II clinical status. It was analysed by using (SwissADME server http://www.swissadme.ch/). The molecular weight needs to be ≤ 500, whereas the hydrogen (H) bond acceptance number needs to be ≤ 10 and the hydrogen (H) bond donor number needs to be ≤ 5. Lipinski rule of 5 (octanol–water partition) log P value of drug like compound needs to be ≤ 5. [60].

## Results

### GC-MS assessment and determination of compounds of interest

Cedrene (M/Z 204, RRT 26.7 mins, 11.627%) was detected as the most abundant compound in *A. scorparia* EO, whereas cedrol (M/Z 222, RRT 8.4 mins, 13.487%) was the major compound by occurrence in *T. orientalis* EO. Meanwhile, digoxigenin (M/Z 388, RRT 2.68 mins, 4.178%) and lauric acid (M/Z 200, RRT 9.32 mins, 0.769%) were detected in the most bioactive TLC isolated fraction of *C. bracteosus,* and were the only phytochemical compounds present in the corresponding fraction (Fig 1). The complete GC-MS profile of *C. bracteosus* has been summarised in S1 Table, while S2 Table shows the profiles for *A. scoparia* and *T. orientalis.*

### Physicochemical properties of NiV and hMPV proteins

By the application ProtParam, the physicochemical properties were analysed. Table 1 shows the total molecular weight of positive and negative amino acid residues were 67039.03 and 57357.27 for NiV attachment glycoprotein (G) and for NiV fusion glycoprotein (F0) whereas the isoelectric point (pI) value indicates acidic nature (5.84) of attachment glycoprotein

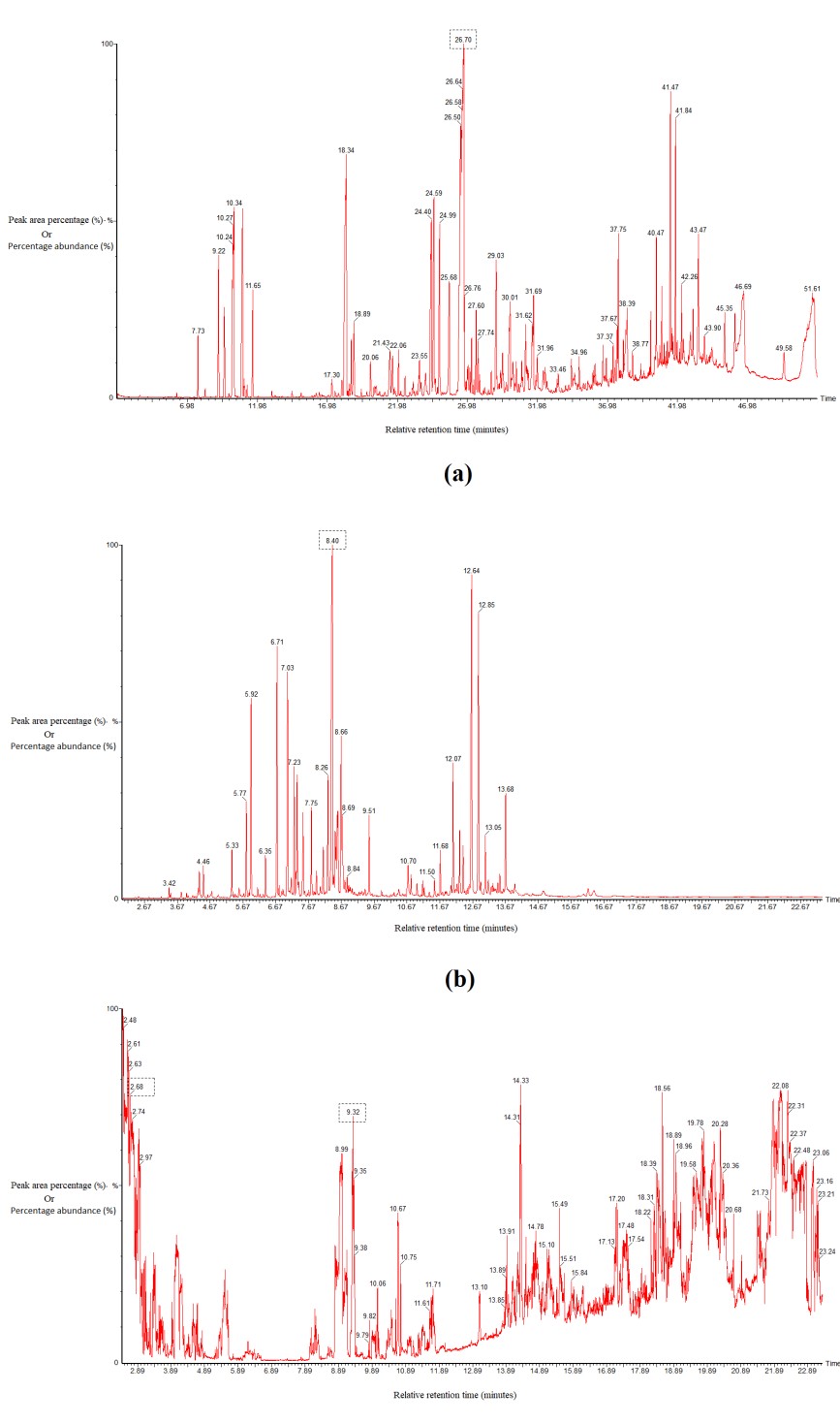

**Fig 1. GC-MS chromatograms of *A. scoparia* EO indicating cedrene (a).** *T. orientalis* EO indicating cedrol (b) and *C. bracteosus* chloroform bark fraction indicating digoxigenin and lauric acid (c).

**Table 1. Physicochemical properties of NiV attachment glycoprotein (G), NiV fusion glycoprotein (F0) and hMPV fusion glycoprotein (F0).**

| Physicochemical property | NiV attachment glycoprotein (G) | NiV fusion glycoprotein (F0) | hMPV fusion glycoprotein (F0) |
|---|---|---|---|
| Molecular weight | 67039.03 | 57357.27 | 2589.34 |
| Number of amino acids | 602 | 520 | 24 |
| Negatively charged residues | 53 | 47 | 0 |
| Positively charged residues | 61 | 44 | 0 |
| pI | 8.58 | 5.84 | 5.52 |
| Ext. coefficient 1 | 76750 | 40395 | None |
| Ext. coefficient 2 | 75750 | 39770 | None |
| Instability Index | 34.56 | 36.94 | 22.10 |
| Aliphatic Index | 90.95 | 109.63 | 231.25 |
| GRAVY | −0.178 | 0.130 | 2.921 |

(G), whereas the fusion glycoprotein (F0) shows basic nature with an indicative pI of 8.58. Meanwhile, the total molecular weight of positive and negative amino acid residues for hMPV fusion glycoprotein (F0) was 2589.34. Whereas the isoelectric point (pI) value of 5.52 indicates an acidic nature. The low value of GRAVY for the selected receptors of NiV and hMPV suggests significant interactions with water molecules and values for aliphatic indexes and instability index, which determines the protein stability for docking.

## Docking assessment

The results of this *in silico* study revealed the potential interactions between chloroquine, cedrene, cedrol, digoxigenin, lauric acid and attachment glycoprotein (G) and fusion glycoprotein (F0) of NiV. The higher docking energy score was used to determine the docking strength of each ligand molecule with its respective receptor. Using Autodock Vina software, the cut-off value for predicting high docking energy between a ligand and a receptor has been set at −6 kcal/mol in previous studies [61,62].

Digoxigenin indicated the highest binding affinities for both NiV attachment glycoprotein (G) and fusion glycoprotein (F0) at −8.3 kcal/mol and −7.7 kcal/mol respectively, followed by cedrene (−6.3 kcal/mol), cedrol (−6.2 kcal/mol), and lauric acid (−4.1 kcal/mol) pronounced as the lowest with their docking with the NiV fusion glycoprotein (F0). Meanwhile, these phytochemicals indicated binding affinities of −7.2 kcal/mol for cedrol, −7.1 kcal/mol for cedrene and −4.9 kcal/mol for lauric acid, which again pronounced as the lowest among the set when docked with the NiV attachment glycoprotein (G). Intriguingly, the binding affinities of all the docked compounds were higher with NiV attachment glycoprotein (G) than NiV fusion glycoprotein (F0). Chloroquine also showed relatively good binding affinities of −6.1 kcal/mol when docked with NiV attachment glycoprotein (G) and −5.5 kcal/mol upon docking with NiV fusion glycoprotein (F0). Remarkably, the binding affinities of digoxigenin, cedrene and cedrol even surpassed that of the drug control chloroquine. Figs 2–4 elucidate the analogy of binding sites, intermolecular bonds of docked receptor-ligand complexes and their amino acid residues that are within the proximity of 3.6 Å, assumptively considered for strong interactions [63].

Once more, digoxigenin indicated the highest binding affinity for hMPV fusion glycoprotein (F0) at −8.1 kcal/mol, followed by cedrene (−6.6 kcal/mol), cedrol (−6.4 kcal/mol), and lauric acid (−4.3 kcal/mol) pronounced as the lowest with their docking with the hMPV fusion glycoprotein (F0). Meanwhile, chloroquine indicated a binding affinity of −5.8 kcal/mol when docked with hMPV fusion glycoprotein (F0). The binding affinities of digoxigenin, cedrene and cedrol exponentially surpassed that of the drug control chloroquine.

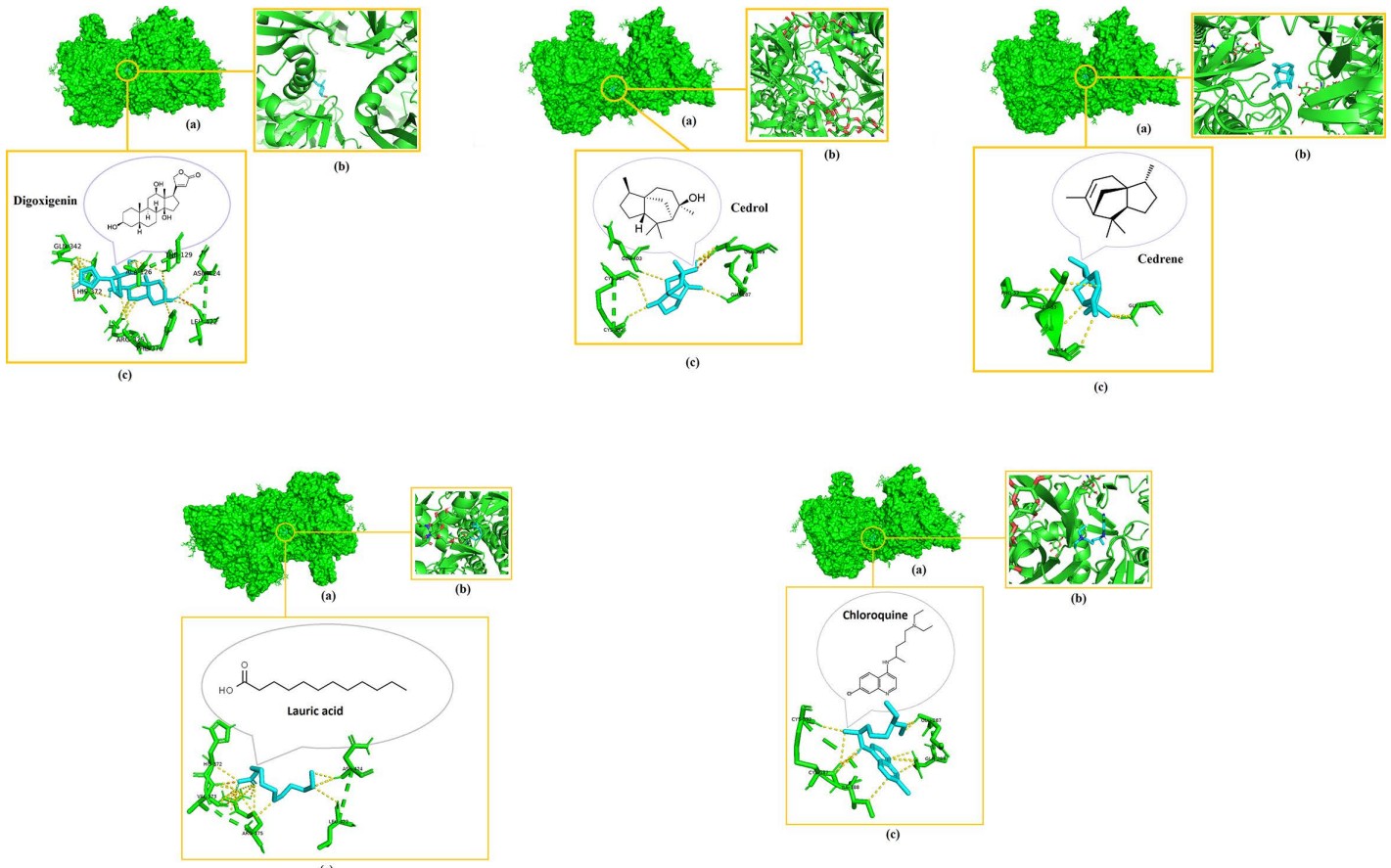

**Fig 2. 3D visualization of the molecular interactions after docking of digoxigenin, cedrol and cedrene with NiV fusion glycoprotein (F0).** (a) Indicates binding domain, (b) Binding site and (c) Interacting amino acids of receptor (green) and ligand (blue) that are withing the proximity of 3.6 Å.

## Predictive analysis for drug likeness, toxicity and ADME

The drug likeness of the compounds was performed by Lipinski Rule of Five, which has been referred to as drugs that were orally administered. A compound that follows three out of the five rules was predicted as a therapeutic drug that has succeeded in *in-silico* studies [64]. Table 2 indicates that all ligands selected obey this rule, with cedrene being the most obedient. Table 3 shows the results of toxicity prediction analysis and pharmacokinetic properties depicting ADME are summarised in Table 4. According to PkCSM toxicity endpoints for therapeutic drug candidates [65], the selected ligands indicate a lower value (< 0.477 log mg/kg/day) for maximum human tolerance, where the tolerance value was less than the control drug chloroquine. All four ligands showed *T. pyriformis* toxicity, which was greater than −0.5 log ug/L with digoxigenin being the least toxic. Meanwhile, acute toxicity was significantly less towards flathead minnows. Moreover, high values indicate that these ligands are readily absorbed through the human small intestine. Aside from digoxigenin, three out of four ligands tested showed higher skin permeability than −2.5 log Kp. Meanwhile, the steady ate volume of distribution in humans (VDss) was low for digoxigenin and was the lowest out of the four ligands tested. All four ligands indicated the ability to readily cross the blood-brain barrier (BBB) since the values are greater than 0.3 log BB with cedrol being the most permeable. However, in case of central nervous system (CNS) permeability, only cedrene has shown the ability to penetrate the CNS. Interestingly, all four ligands tested showed high permeability towards Caco-2 cell line

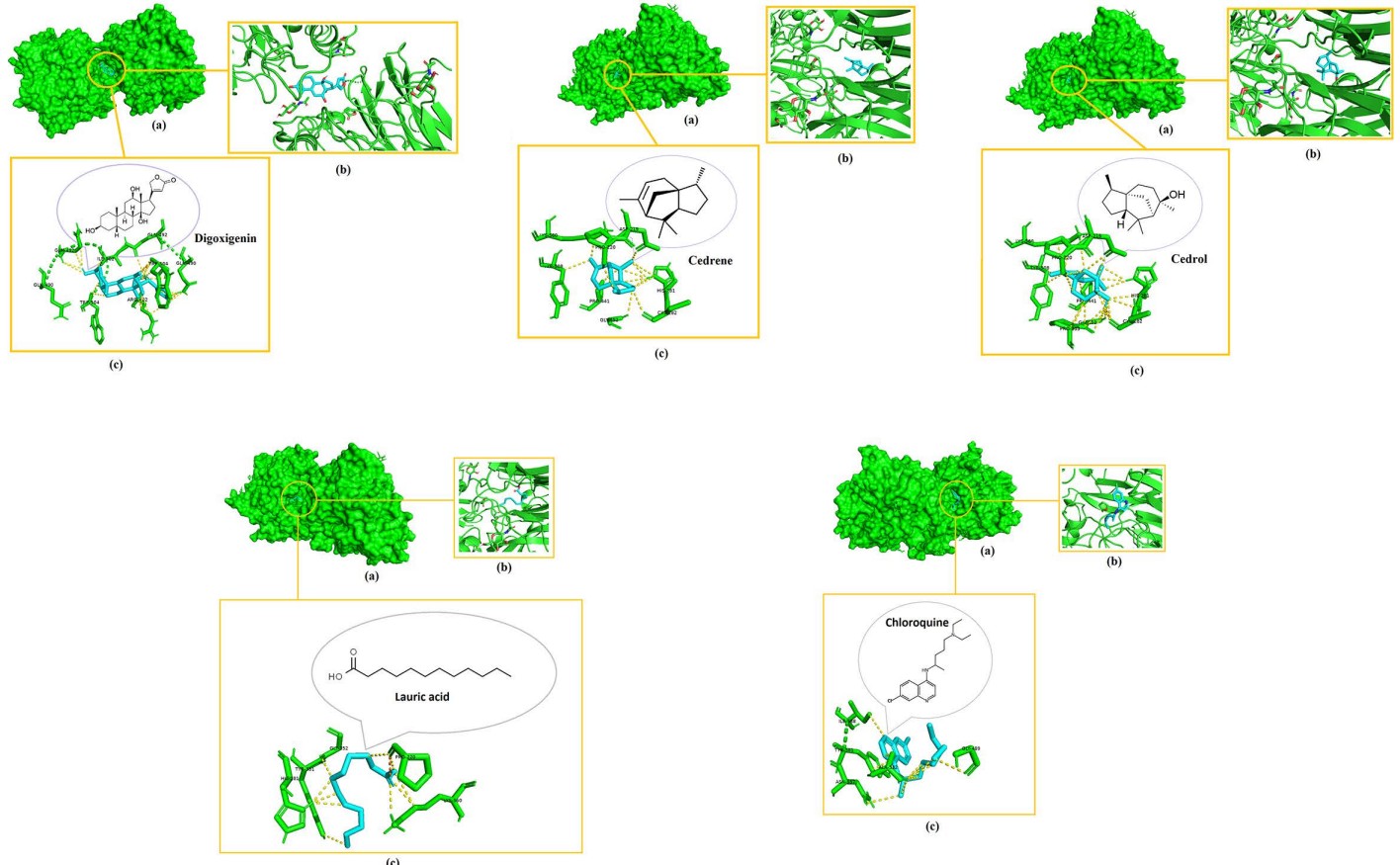

**Fig 3. 3D visualization of the molecular interactions after docking of digoxigenin, cedrene and cedrol with NiV attachment glycoprotein (G).** (a) Indicates binding domain, (b) Binding site and (c) Interacting amino acids of receptor (green) and ligand (blue) that are withing the proximity of 3.6 Å.

(>0.9 log Papp in 10−6 cm/s), which are associated with human epithelial colorectal adenocarcinoma, where cedrol has the most Caco-2 cell penetrating ability. Digoxigenin appears to affect the P-glycoprotein substrate, which involves the extrusion of toxins and xenobiotics from cells. Meanwhile, all four selected ligands were affecting the CYP3A4 substrate, which are cytochrome P450 enzymes that metabolise drugs. Subsequently, cedrol acted as an inhibitor of CYP1A2 and CYP2C19 substrates, which are also part of cytochrome P450 enzymes. In comparison to the reference drug chloroquine, digoxigenin, cedrene and cedrol showed adequate characteristics that can be potentially beneficial to qualify as safe antiviral agents, since they did not indicate AMES toxicity, which shows that compounds are mutagenic and can act as a potential carcinogen. Also, unlike chloroquine, they were not hERG inhibitors that can lead to the development of acquired long QT syndrome.

## Analysis of molecular dynamics simulations

Protein conformational dynamics are the most imperative aspect of its function. The functional information of a protein molecule was preserved in its structure. The structures are required to be utterly examined to comprehend their functional variability [66]. MD simulation using GROMACS in conjunction with AMBER was utilised in this study to investigate the conformational components of viral protein receptor–ligand interactions. RMSD and RMSF depicts the stability analysis and atom dynamics of a docked protein during a 100 ns run time for both complexes. RMSD and RMSF values

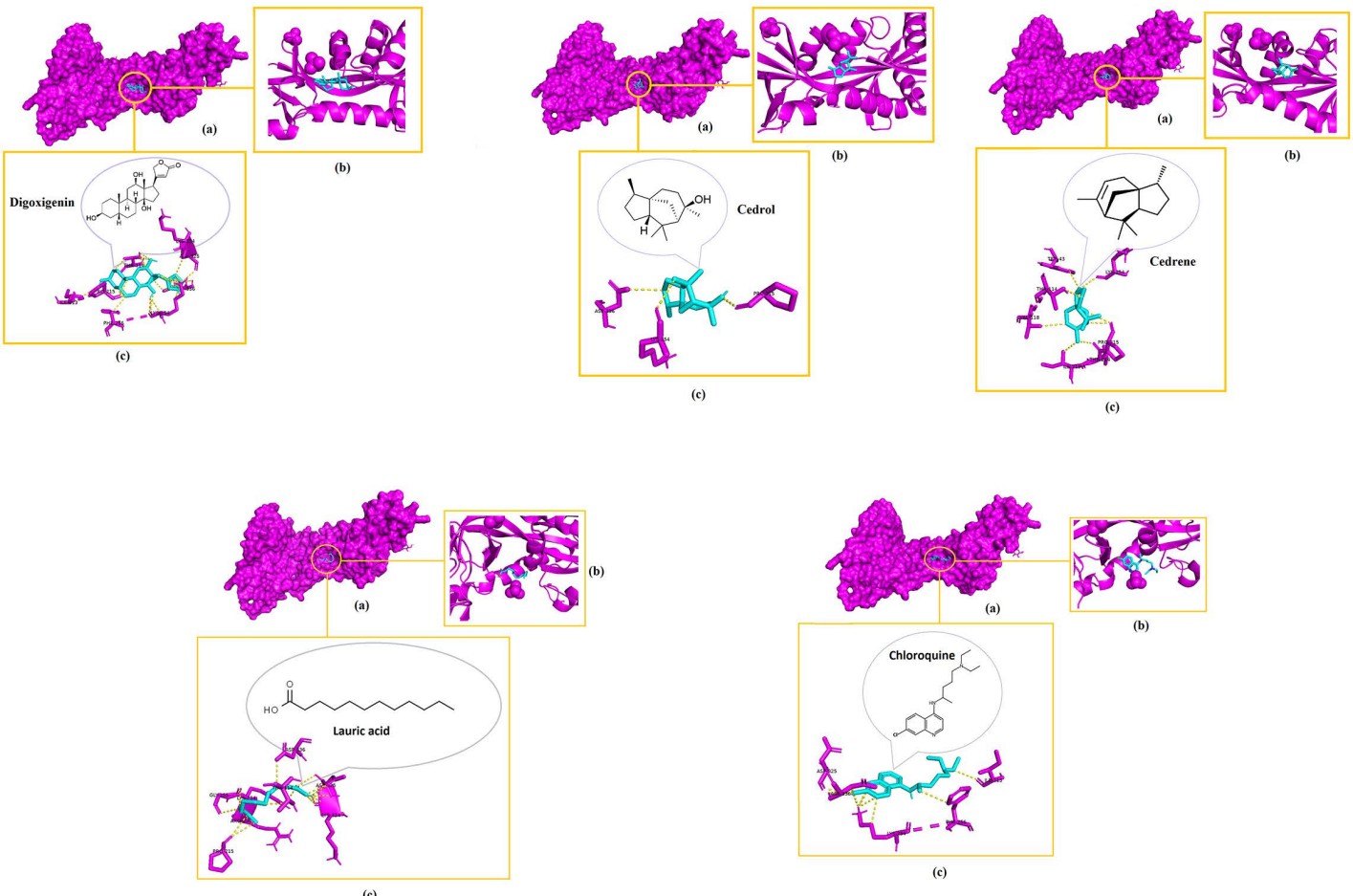

**Fig 4. 3D visualization of the molecular interactions after docking of digoxigenin, cedrol and cedrene with hMPV fusion glycoprotein (F0).** (a) Indicates binding domain, (b) Binding site and (c) Interacting amino acids of receptor (magenta) and ligand (blue) that are withing the proximity of 3.6 Å.

**Table 2. Drug likeness analysis of the selected ligands.**

| Ligand | Formula | Molecular weight | #H-bond acceptors | #H-bond donors | Log P-value |
|---|---|---|---|---|---|
| Digoxigenin | C23H34O5 | 390.51 | 5 | 3 | 2.58 |
| Cedrene | C15H24 | 204.35 | 0 | 0 | 4.42 |
| Cedrol | C15H26O | 222.37 | 1 | 1 | 3.61 |
| Chloroquine (reference drug) | C18H26ClN3 | 319.87 | 2 | 1 | 4.62 |

of protein-ligand complexes were monitored to identify any minor or major alterations in the structural configuration of the viral protein receptor after ligand binding. A lower RMSD value has been proposed as an indicator of a more stable docked configuration [67]. The data for RMSD and RMSF of selected ligand-receptor complexes were summarised in Table 5 and Figs 5-6. Comparing the behaviour of the 2VWD (A+B chain) simulations, all ligands maintained a stable structure similar to the free protein with no significant differences. Note only the RMSD of the Cedrene+2VWD complex, which showed a higher value. For the 5EVM target (A+B+C+D+E+F chains), all ligands produced better protein

 

**Table 3. pkCSM toxicity properties of the selected ligands.**

| Toxicity property model name | Ligand with predicted value and unit of measurement | | | |
|---|---|---|---|---|
| | Digoxigenin | Cedrene | Cedrol | Chloroquine (reference drug) |
| AMES toxicity | No | No | No | Yes |
| Max. tolerated dose (human) | −0.733 log mg/kg/day | 0.004 log mg/kg/day | −0.036 log mg/kg/day | 0.14 log mg/kg/day |
| hERG I inhibitor | No | No | No | No |
| hERG II inhibitor | No | No | No | Yes |
| Oral Rat Acute Toxicity (LD50) | 2.593 mol/kg | 1.579 mol/kg | 1.666 mol/kg | 2.818 mol/kg |
| Oral Rat Chronic Toxicity (LOAEL) | 1.9 log mg/kg bw/day | 1.374 log mg/kg bw/day | 1.254 log mg/kg bw/day | 0.924 log mg/kg bw/day |
| Hepatotoxicity | No | No | No | Yes |
| Skin Sensitisation | No | No | Yes | No |
| *T. pyriformis* toxicity | 0.365 log ug/L | 1.433 log ug/L | 1.363 log ug/L | 1.346 log ug/L |
| Minnow toxicity | 1.284 log mM | 0.246 log mM | 0.851 log mM | 0.559 log mM |

**Table 4. pkCSM pharmacokinetic properties of the selected ligands.**

| Pharmacokinetic property | | Ligand with predicted value | | | | Unit of measurement |
|---|---|---|---|---|---|---|
| Property | Model name | Digoxigenin | Cedrene | Cedrol | Chloroquine (reference drug) | |
| Absorption | Water solubility | -4.299 | -5.829 | -4.361 | -3.84 | log mol/L |
| | Caco-2 permeability (H) | 1.107 | 1.407 | 1.496 | 1.471 | log Papp in 10-6 cm/s |
| | Intestinal absorption (human) | 95.714 | 96.112 | 93.844 | 88.665 | % Absorbed |
| | Skin Permeability | -3.872 | -1.797 | -2.163 | -2.764 | log Kp |
| | P-glycoprotein substrate | Yes | No | No | Yes | - |
| | P-glycoprotein I inhibitor | No | No | No | No | - |
| | P-glycoprotein II inhibitor | No | No | No | No | - |
| Distribution | VDss (human) | -0.106 | 0.77 | 0.572 | 1.371 | log L/kg |
| | Fraction unbound (human) | 0.181 | 0.154 | 0.249 | 0.185 | Fu |
| | BBB permeability | -0.694 | 0.154 | 0.627 | 0.382 | log BB |
| | CNS permeability | -3.384 | -1.799 | -2.216 | -2.336 | log PS |
| Property | Model name | Digoxigenin | | | | |
| Metabolism | CYP2D6 substrate | No | No | No | Yes | - |
| | CYP3A4 substrate | Yes | Yes | Yes | Yes | - |
| | CYP1A2 inhibitor | No | No | Yes | No | - |
| | CYP2C19 inhibitor | No | No | Yes | No | - |
| | CYP2C9 inhibitor | No | No | No | No | - |
| | CYP2D6 inhibitor | No | No | No | No | - |
| | CYP3A4 inhibitor | No | No | No | No | - |
| Excretion | Total Clearance | 0.605 | 0.929 | 0.837 | 1.072 | log ml/min/kg |
| | Renal OCT2 substrate | No | No | No | Yes | - |

stabilisation compared to the free protein simulation. The same occurred with the 5WB0 target. All ligands produced better target stability when compared to the free structure.

The data for RMSD and RMSF of selected ligand-receptor complexes were summarised in Table 5 and Figs 5-6. In case of 5WB0, all ligand molecules maintained consistent protein stability throughout RMSD evaluation. The RMSD of all

**Table 5. RMSD and RMSF data for the selected ligand-receptor complexes.**

| Ligand + Receptor complex | Chain | Mean RMSD | Highest RMSD peak | Mean RMSF | Highest RMSF peak | Rg (nm) | Mean SASA (nm²) |
|---|---|---|---|---|---|---|---|
| Cedrene + 2VWD | A* | 0.2504 | 0.315531 | 0.1291 | 0.4697 | 2.114546505 | 352.3124367 |
| | B | 0.2520 | 0.303348 | 0.1164 | 0.429 | 2.102585293 | |
| Cedrol + 2VWD | A | 0.2251 | 0.270554 | 0.1197 | 0.4156 | 2.115024539 | 347.7997839 |
| | B* | 0.2672 | 0.318185 | 0.1282 | 0.4817 | 2.107862754 | |
| Digoxigenin + 2VWD | A* | 0.2135 | 0.269364 | 0.1184 | 0.5551 | 2.10920534 | 357.7350688 |
| | B* | 0.2422 | 0.302281 | 0.1304 | 0.5431 | 2.111888603 | |
| Chloroquine + 2VWD | A | 0.22959543 | 0.2816858 | 0.5522 | 0.115179661 | 2.099142004 | 345.7522558 |
| | B | 0.238943256 | 0.2920055 | 0.115179661 | 0.5522 | 2.102723035 | |
| Apo-2VWD | A | 0.22959543 | 0.2816858 | 0.119782809 | 0.472 | 2.103010288 | 347.6891768 |
| | B | 0.222768206 | 0.2803406 | 0.119164634 | 0.5673 | 2.102385522 | |
| Cedrene + 5EVM | A | 0.2981 | 0.394917 | 0.1301 | 0.562 | 3.193766515 | 1094.919501 |
| | B* | 0.2761 | 0.395697 | 0.1247 | 0.7121 | 3.199306922 | |
| | C | 0.2616 | 0.339315 | 0.1278 | 0.5871 | 3.194962727 | |
| | D | 0.2517 | 0.3024308 | 0.1346 | 0.553 | 3.199821661 | |
| | E | 0.2562 | 0.352234 | 0.1379 | 0.5561 | 3.196453023 | |
| | F* | 0.2531 | 0.31381 | 0.1364 | 0.7229 | 3.197754842 | |
| Cedrol + 5EVM | A | 0.2705 | 0.329855 | 0.1407 | 0.6095 | 3.193624838 | 1095.72 |
| | B | 0.2737 | 0.343382 | 0.1425 | 0.5961 | 3.185824514 | |
| | C* | 0.2687 | 0.36837 | 0.1373 | 0.9987 | 3.203840897 | |
| | D* | 0.3137 | 0.382505 | 0.1342 | 0.9087 | 3.167391093 | |
| | E | 0.3005 | 0.3821838 | 0.1316 | 0.7136 | 3.163999615 | |
| | F | 0.2527 | 0.315537 | 0.1246 | 0.7399 | 3.177119935 | |
| Digoxigenin + 5EVM | A | 0.2595 | 0.330201 | 0.1300 | 0.6897 | 3.170009753 | 1096.478089 |
| | B | 0.2593 | 0.316859 | 0.1323 | 0.7421 | 3.175608381 | |
| | C* | 0.2410 | 0.315383 | 0.1292 | 0.8145 | 3.156583703 | |
| | D | 0.3084 | 0.391474 | 0.1423 | 0.6859 | 3.208604667 | |
| | E* | 0.3261 | 0.461621 | 0.1467 | 0.8119 | 3.227074492 | |
| | F | 0.2782 | 0.362017 | 0.1309 | 0.6649 | 3.191858868 | |
| Chloroquine + 5EVM | A | 0.277493049 | 0.354657 | 0.137890132 | 0.555 | 3.192640723 | 1097.843364 |
| | B | 0.25235077 | 0.317403 | 0.136374342 | 0.566 | 3.183398137 | |
| | C | 0.232889857 | 0.308868 | 0.126101333 | 0.5868 | 3.184527996 | |
| | D* | 0.303384705 | 0.399293 | 0.136682456 | 0.7615 | 3.171065708 | |
| | E* | 0.247762069 | 0.300198 | 0.13021557 | 0.7902 | 3.185875497 | |
| | F | 0.239761245 | 0.320151 | 0.125246667 | 0.6199 | 3.191397125 | |
| Apo-5EVM | A | 0.375596375 | 0.435579 | 0.137890132 | 0.555 | 3.1636667 | 1091.106737 |
| | B | 0.313480649 | 0.458096 | 0.136374342 | 0.566 | 3.206978921 | |
| | C* | 0.320199452 | 0.474533 | 0.126101333 | 0.5868 | 3.189789641 | |
| | D | 0.312994302 | 0.392103 | 0.136682456 | 0.7615 | 3.172152513 | |
| | E* | 0.352424374 | 0.449767 | 0.13021557 | 0.7902 | 3.15213117 | |
| | F | 0.375987187 | 0.496797 | 0.125246667 | 0.6199 | 3.190284423 | |
| Cedrene + 5WB0 | NA | 0.3912 | 0.543764 | 0.2106 | 0.9901 | 2.910784907 | 228.0793689 |
| Cedrol + 5WB0 | NA | 0.4010 | 0.605333 | 0.2206 | 0.8321 | 3.030316217 | 231.2327319 |
| Digoxigenin + 5WB0 | NA | 0.3966 | 0.601756 | 0.1972 | 0.7845 | 2.924356596 | 228.0793689 |

*(Continued)*

**Table 5.** (Continued)

| Ligand + Receptor complex | Chain | Mean RMSD | Highest RMSD peak | Mean RMSF | Highest RMSF peak | Rg (nm) | Mean SASA (nm²) |
|---|---|---|---|---|---|---|---|
| Chloroquine + 5WB0 | NA | 0.476526106 | 0.706765 | 0.247967841 | 0.6697 | 2.917000043 | 229.178371 |
| Apo-5WB0 | NA | 0.527516448 | 0.828715 | 0.257354846 | 1.1873 | 2.900731414 | 227.9265142 |

NA: Not Applicable, *: Chain with the highest RMSD/RMSF peak, 2VWD: NiV attachment glycoprotein (G), 5EVM: NiV fusion glycoprotein (F0), 5WB0: hMPV fusion glycoprotein (F0), Radius of gyration (Rg), Solvent accessible surface area (SASA).

complexes with ligands remain around 0.15-0.3 nm without significant fluctuations till 100 ns. However, major fluctuations were detected RMSF analysis between residues 200 and 600, where the residues remained steady around 0.2 nm. Protein 5EVM and its ligand complexes showed minor fluctuations in their RMSDs between 0–20 ns and became stable till 60 ns. Once more, minor fluctuations were observed between 60–70 ns and became stable onwards. The RMSF of most residues remained stable around 0.5 nm up to 440 residues with few minor fluctuations in around 100, 160 and 260 residues. Furthermore, all ligand molecules provide protein-like stability with 5WB0. The RMSD of all complexes with ligands remain around 0.4 nm with a slight oscillation at the end of 100 ns for cedrol. In the case of digoxigenin, minor fluctuations were observed between 20 ns and 30 ns, and between 60 ns and 70 ns, after which they became stable. Major fluctuations were observed for RMSF between 90 and 100 residues, with the largest oscillation shown for cedrene in this region, at approximately 1 nm. Subsequently, minor fluctuations between 250 and 300 residues were observed for 5WB0 and its ligand complexes.

Notably, ligand-receptor complexes such as digoxigenin + 2VWD, cedrene + 5EVM, cedrene + 5WB0, and cedrol + 5WB0 exhibited RMSD values that were on par and in some cases, lower than the apoprotein (≤ 0.2-0.3 nm). The temporal plot of the RMSD and RMSF values for the carbon backbone indicates that most complexes exhibit fluctuations of less than 0.3 nm, as illustrated in Figs 5-6. Although there was a notable deviation, the RMSD values for the apoprotein complex maintained a stable level until the conclusion of the simulation. The results imply that most proteins and ligands remained stable within their complexes during the entire simulation.

## SASA and Rg analysis

We assessed the SASA of residues associated with significant drug-target interactions or catalytic functions, due to compelling results from the RMSD and RMSF analyses. When comparing ligand-receptor complex systems, the SASA profile (Table 5 and Fig 7) distinctly indicates a decrease in the SASA of key residues. The enzymatic functions of the targeted proteins are hindered in both systems because of the restricted accessibility to essential residues within the complex systems, thereby lowering the chances of complex interactions. Based on the SASA analysis, this has occurred during the simulation time intervals at a timescale of 100 ns (10,000 ps). The average values recorded for each complex system are shown in Table 5, which suggests possible dynamics and stability of the system. Furthermore, the average SASA values ranged between 347.6-357.7 nm² for 2VWD, 1097.8-1097.8 nm² for 5EVM, and 227.9-231.2 nm² for 5WB0 ligand-receptor complexes and their apo proteins. However, the most stable ligand-receptor complex appears to be cedrol + 2VWD with a SASA of 347.79 nm², which was closest to that of its apo protein with a SASA of 347.68 nm².

The radius of gyration (Rg) serves as an indicator of a structure's compactness. A lower degree of fluctuation and consistency throughout the simulation suggests that the system exhibits greater compactness and stiffness. To evaluate the compactness of each complex, we examined the Rg values for the following complexes: cedrol + 2VWD, cedrene + 2VWD, digoxigenin + 2VWD, chloroquine + 2VWD, Apo-2VWD, cedrol + 5EVM, cedrene + 5EVM, digoxigenin + 5EVM, chloroquine + 2VWD, Apo-2VWD, cedrol + 5WB0, cedrene + 5WB0, digoxigenin + 5WB0, chloroquine + 5WB0, Apo-5WB0 with average Rg values ranging from 2.10-2.11 nm for 2VWD complexes, 3.15-3.22 nm for 5EVM complexes, and 2.9-3 nm for

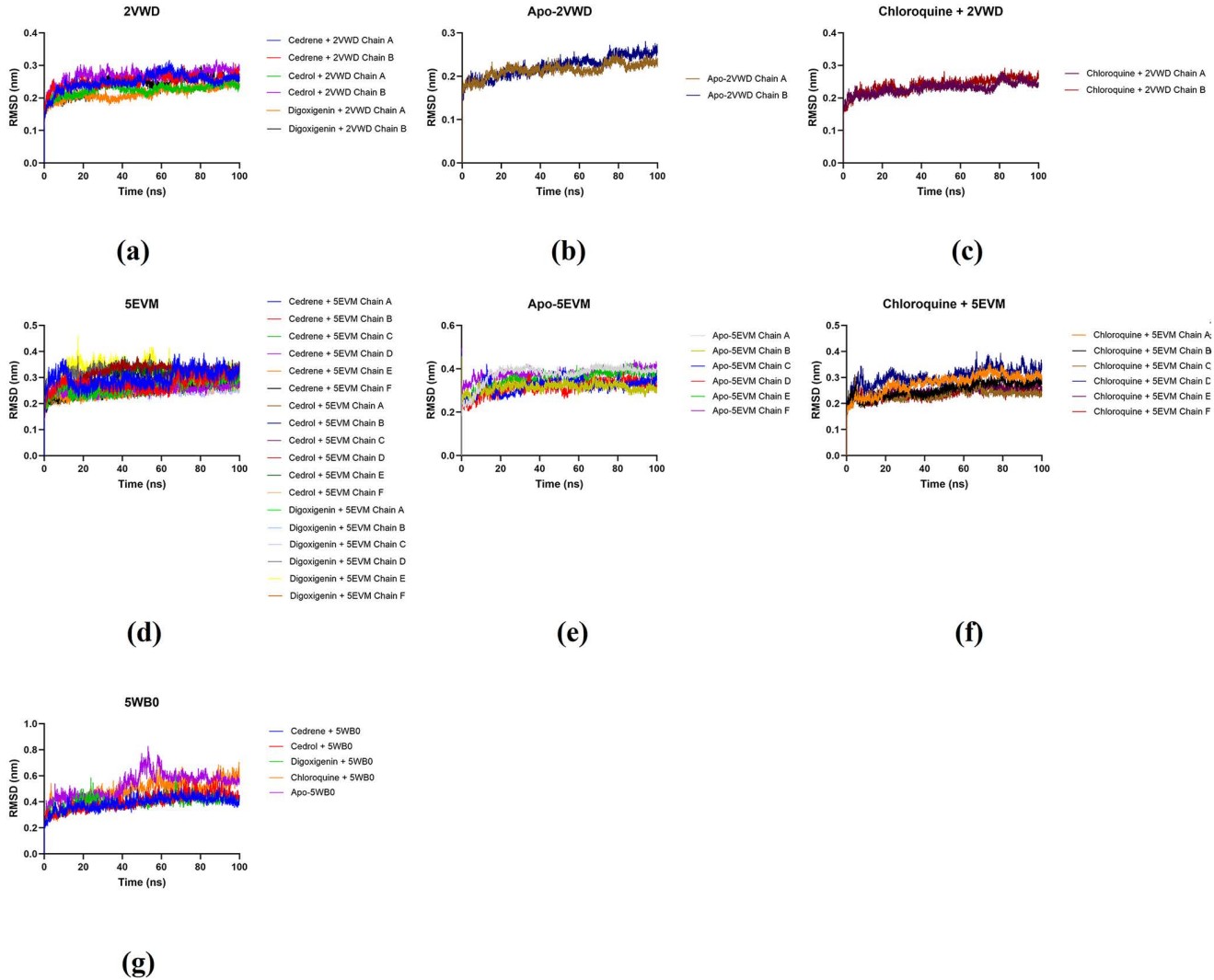

**Fig 5. Molecular dynamics simulation based on RMSDs of receptor-phytochemical ligand complexes of 2VWD:** NiV attachment glycoprotein G (a), apo-protein 2VWD (b), receptor-chloroquine ligand complex of 2VWD (c) receptor-phytochemical ligand complexes of 5EVM: NiV fusion glycoprotein F0 (d), apo-protein 5EVM (e), and receptor-chloroquine ligand complex of 5EVM (f) receptor-phytochemical ligand complexes of 5WB0: hMPV fusion glycoprotein F0 (g).

5WB0 complexes and their apo proteins, respectively (Table 5 and Fig 7). The digoxigenin + 5EVM complex displayed the highest Rg value at 3.15 nm, whereas the cedrene + 2VWD complex recorded the lowest Rg value at 2.1. All complexes demonstrated consistent fluctuations, indicating greater compactness and increased rigidity.

## Hydrogen bond analysis

Hydrogen bonds play a pivotal role in defining the stability of simulation systems, acting as a cornerstone of structural integrity. This insight was gleaned through the application of AMBER16 tools, meticulously analysing each frame across various time intervals. Remarkably, 2VWD, 5EVM and 5WB0 complex protein receptor systems revealed robust and sound hydrogen bonding with ligands cedrol, digoxigenin, lauric acid and chloroquine. NiV attachment

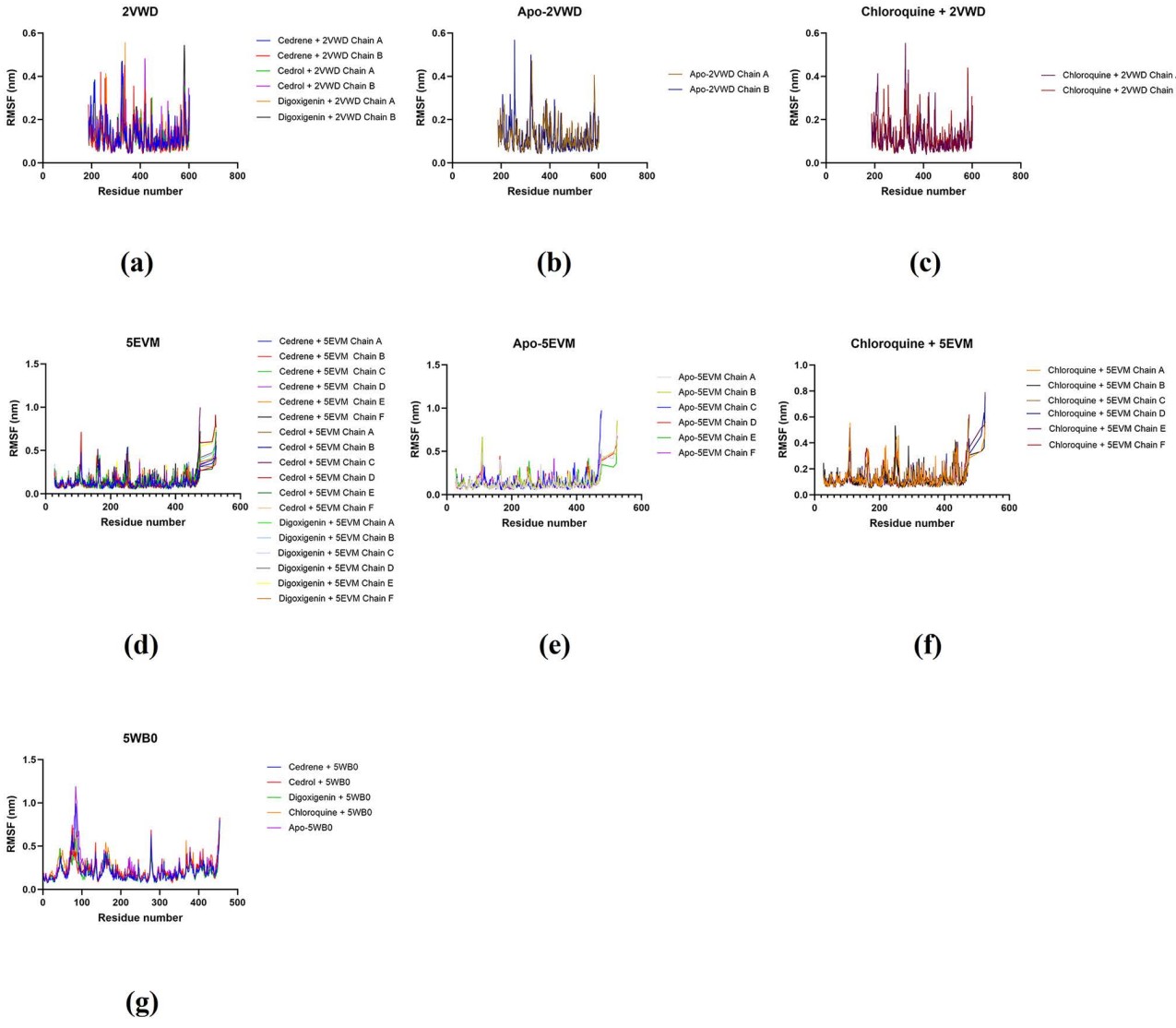

**Fig 6. Molecular dynamics simulation based on RMSFs of receptor-phytochemical ligand complexes of 2VWD:** NiV attachment glycoprotein G (a), apo-protein 2VWD (b), receptor-chloroquine ligand complex of 2VWD (c) receptor-phytochemical ligand complexes of 5EVM: NiV fusion glycoprotein F0 (d), apo-protein 5EVM (e), and receptor-chloroquine ligand complex of 5EVM (f) receptor-phytochemical ligand complexes of 5WB0: hMPV fusion glycoprotein F0 (g).

glycoprotein (G) (PDB ID: 2VWD), showed hydrogen bonding between the O atom of CYS282 of the receptor and the O atom of cedrol, whereas between the NE2 atom of GLN 490 (A) of the receptor and the O5 atom of digoxigenin, followed by between the O atom of PRO220 and the O1 atom of lauric acid. Meanwhile, NiV fusion glycoprotein (F0) (PDB ID: 5EVM) showed hydrogen bonding between the NE2 atom of GLN289 of the receptor and the O atom of cedrol, whereas between the NE2 atom of GLN342 together with the N atom of VAL373 of the receptor and O4 and O1 atoms of digoxigenin respectively, followed by between O and N atoms of VAL373 of the receptor and the O1 atom of lauric acid, furthermore between the O atom of CYS387 of the receptor and the N2 atom of chloroquine. Moreover, hMPV fusion glycoprotein (F0) (PDB ID: 5WB0) showed hydrogen bonding between the O atom of PRO215 of the

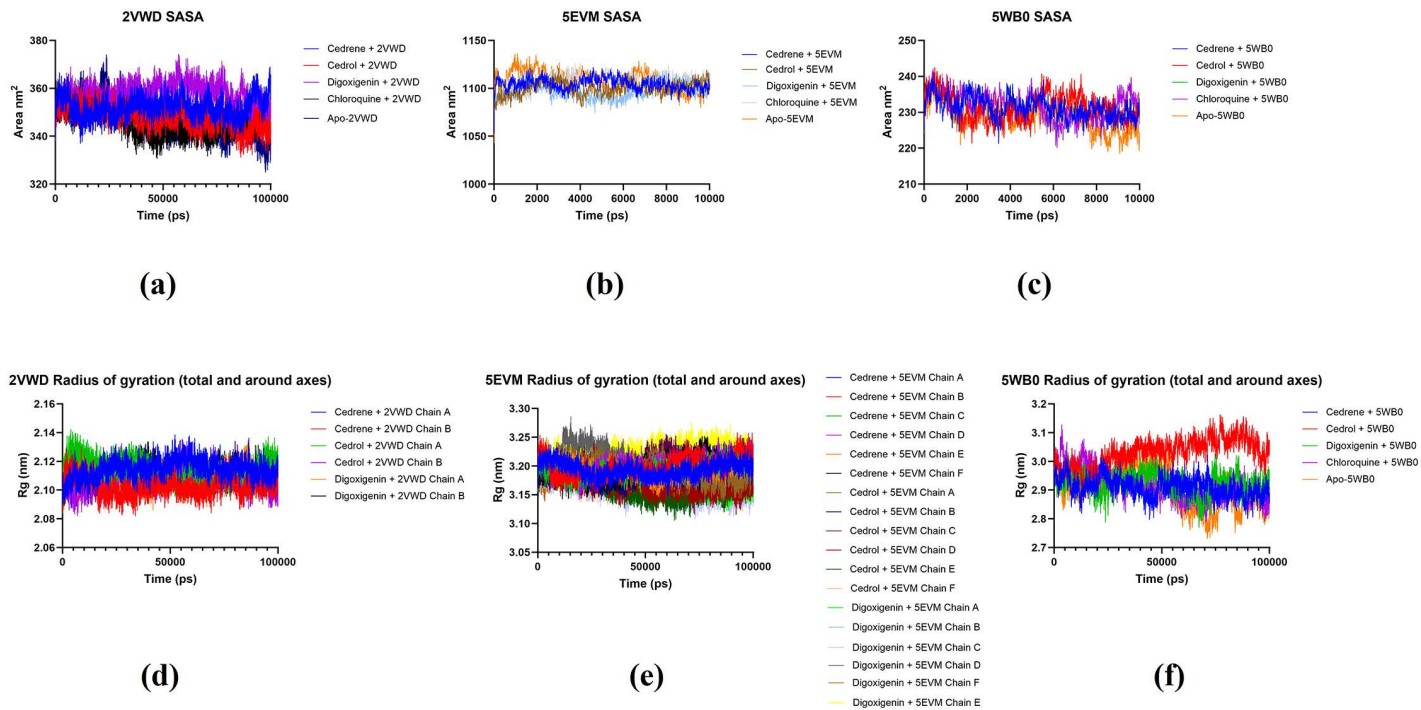

**Fig 7. Molecular dynamics simulation based on SASAs of receptor-ligand complexes of 2VWD:** NiV attachment glycoprotein G (a), 5EVM: NiV fusion glycoprotein F0 (b), apo-protein 5EVM (e), and 5WB0: hMPV fusion glycoprotein F0 (g). Followed by Rgs of receptor-ligand complexes of 2VWD: NiV attachment glycoprotein G (d), 5EVM: NiV fusion glycoprotein F0 (e), apo-protein 5EVM (e), and 5WB0: hMPV fusion glycoprotein F0 (f).

receptor and the O atom of cedrol, whereas between the O atom of LYS254 of the receptor and the O1 atom of digoxigenin, followed by between the OD1 atom of ASP325 together with the O atom of LYS324 of the receptor and the O1 atom of lauric acid. A bond that persisted with admirable consistency for up to 100 ns, occasionally experiencing only fleeting interruptions. This intricate network of bonds underscores the delicate yet powerful interplay that contributes to the intricate dance of molecular stability and interaction. The ligand cedrene on the other hand, did not show any hydrogen bonds. The nature of hydrogen bonds and their interacting amino acid residues, and hydrophobic interactions of selected ligand-receptor complexes are summarised in Table 6 and Fig 8. Subsequently, the quantity of hydrogen bonds per interacting amino acid residue was assessed throughout the MD simulations to investigate the stability of the hydrogen bonds within protein-ligand complexes. Hence, mapping of the most prominent hydrogen interactions between amino acids and ligands throughout the simulation. (Fig 9).

## Binding free energies analysis

The total binding free energy and stability of the complexes are primarily governed by factors that contribute to the intricate dynamics of drug-target binding. Endpoint free energy modelling techniques, such as MM/PBSA, are renowned for providing far more accurate results than conventional docking binding energy methods. The more negative the values, the more effective the binding free energy [68]. This led us to employ these advanced techniques to delve deeper into the binding free energies of both complexes.

Upon thorough analysis, we discovered that cedrene indicated the lowest average binding free energy for NiV attachment glycoprotein (G) (PDB ID: 2VWD) at −22.9 kcal/mol and, followed by digoxigenin (−19.58 kcal/mol) and cedrol (6.51

**Table 6. Hydrogen and hydrophobic interactions of selected ligand-receptor complexes.**

| Receptor | Ligand | Number of hydrogen bonds | Hydrogen bonding | | Amino acid residues with hydrophobic bonding |
|---|---|---|---|---|---|
| | | | Amino acid residues | Distance (Å) | |
| 2VWD | Cedrene | N/A | N/A | N/A | Cys282, Leu221, Gly352, Pro353, Pro220, Asp219, His281, Tyr351, Pro441, Lys560, Tyr508 |
| | Cedrol | 1 | Cys282 | 2.82 | His281, Gly352, Cys282, Leu221, Tyr351, Asp219, Pro441, Pro353, Pro220, Ile562, Tyr508, Lys560 |
| | Digoxigenin | 2 | Trp504(A) | 2.81 | Trp504, Glu505, Arg402, Gln492, Pro403, Ile502, Gln492, Pro403, Cys503 |
| | | | Gln490(A) | 3.01 | |
| | Lauric acid | 1 | Pro220 | 2.89 | Lys324, Asp336, Asp325, Lys254, Thr114, Phe256, Pro215, Gly255, Val118 |
| | Chloroquine | N/A | N/A | N/A | Gln559, Tyr581, Ile588, Ala558, Gly506, Asn557, Thr531, Gln490, Gly489, Ala532, Pro488 |
| 5EVM | Cedrene | N/A | N/A | N/A | Gly112, Ile426, Pro52, Gln112, Val421, Leu53, Thr54 |
| 5EVM | Cedrol | 1 | Gln289 | 3.19 | Gln287, Cys387, Thr391, Thr286, Cys392, Ser402, Ala400, Ile401 |
| | Digoxigenin | 2 | Gln342 | 3 | Gln342, Val373, Pro374, His372, Arg375, Leu422, Phe376, Gly423 |
| | | | Val373 | 3.19 | |
| | | | Leu422 | 2.96 | |
| | Lauric acid | 1 | Val373 | 3.01 (O1-O), 3.05 (O1-N) | Ile562, Pro220, Tyr508, Lys560, Cys282, Pro353, Gly352, Pro441, Asp219, Phe458, His281, Tyr351 |
| | Chloroquine | 1 | Cys387 | 2.95 | Ile388, Gln403, Gln289, Cys387, Leu285, Glu287, Cys387, Ser402, Thr286, Thr391, Val390, Cys392, Ile284 |
| 5WB0 | Cedrene | N/A | N/A | N/A | Lys254, Ala117, Pro215, Trp43, Thr114, His264, Val118, Thr214, Ile213 |
| 5WB0 | Cedrol | 1 | Pro215 | 2.92 | Pro215, Phe256, Arg253, Ala117, Thr114, Lys324, Lys254, Trp43, Ala113, Asp336 |
| | Digoxigenin | 1 | Lys254 | 3.22 | Ile264, Pro466, Phe260, His264, Arg470, Gly255, Val118, Pro215 |
| | Lauric acid | 2 | Asp325 | 2.84 | Val373, His372, Arg375, Ser371, Val390, Leu422, Phe376, Val425, Asn424, Thr129 |
| | Chloroquine | N/A | N/A | N/A | Lys324, Asp325, Lys254, Asp336, Thr114, Ala117, Pro215, Phe256, Thr214, Ile213 |

NA: Not Applicable, 2VWD: NiV attachment glycoprotein (G), 5EVM: NiV fusion glycoprotein (F0), 5WB0: hMPV fusion glycoprotein (F0), Å: Angstrom.

kcal/mol) pronounced as the highest among the set. Meanwhile, these phytochemicals indicated average binding free energies of −11.48 kcal/mol for cedrol as being the highest, −16.32 kcal/mol for cedrene and −17.55 kcal/mol for digoxigenin, which pronounced as the lowest among the set for NiV fusion glycoprotein (F0) (PDB ID: 5EVM). Subsequently, digoxigenin indicated the lowest average binding free energy for hMPV fusion glycoprotein (F0) (PDB ID: 5WB0) at −17.25 kcal/mol and, followed by cedrol (−9.17 kcal/mol) and cedrene (−4.69 kcal/mol) pronounced as the highest among the set. Chloroquine showed average binding free energies of 11.47 kcal/mol, 36.53 kcal/mol and 14.46 kcal/mol for 2VWD, 5EVM and 5WB0 respectively. Remarkably, the average binding free energies of digoxigenin, cedrene and cedrol were exponentially lower than that of the drug control chloroquine across a time interval of 100 ns, suggesting an enticing level of transient stability during this period. This revelation illuminates the complex's potential, underscoring its significance in the quest for effective drug design.

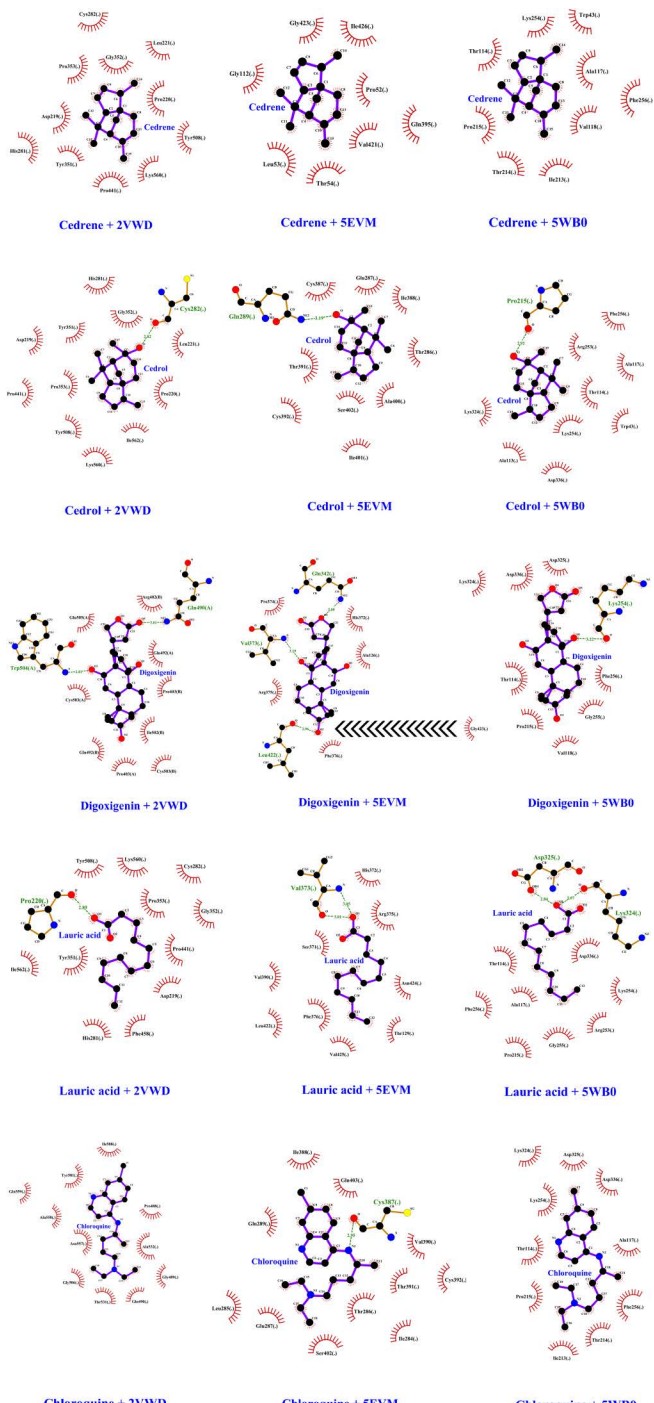

**Fig 8. Ligplot displaying contacts via hydrogen bonds in green dashed lines and hydrophobic interactions in red semi-circles/arcs with radiating lines.**

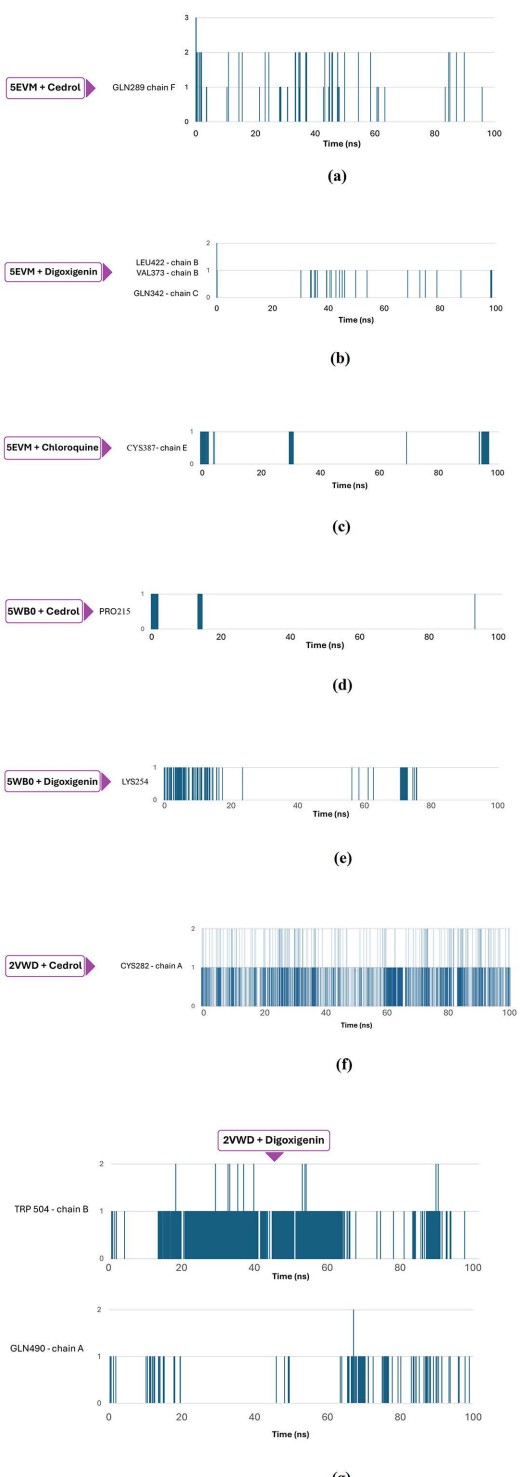

**Fig 9. Hydrogen bond trajectories of receptor-ligand complexes depicting the hot spot residues involved during the simulation time interval of 100 ns.** GLN289 of the F chain of 5EVM generated up to three simultaneous interactions with cedrol in the early stages of the simulation (a). Distribution of hydrogen bonds of digoxigenin combined with the amino acids LEU422, VAL373 of the B chain and GLN342 of the C chain of 5EVM (b). For Chloroquine, it was CYS387 of the E chain that provided the hydrogen bonds with 5EVM (c). Cedrol formed an important hydrogen bond with PRO215 of 5WB0 (d). Digoxigenin formed interactions at the beginning of the simulation with LYS254 of 5WB0 and repeated them around 80 ns (e). Cedrol formed

up to two hydrogen bonds at some points in the simulation with CYS282 of the A chain of 2VWD (f). For digoxigenin, two distinct interactions stand out: that of TRP504 on chain B and GLN490 on chain A of 2VWD (g).

## Discussion

The recent epidemic caused by the virus Nipah in India showed that the current antiviral therapeutic arsenal is inadequate [69,70]. Vaccine development is time-consuming and costly, and large-scale administration can be challenging [71]. Furthermore, the development of antiviral treatments is hindered by the high evolutionary strength of NiV.

The NiV uses the viral G envelope glycoprotein to attach to the host cell receptors ephrin-B2 and/or -B3, followed by the F fusion protein activation, which triggers fusion between the viral envelope G and the host membrane to initiate pathogenesis [72]. These two glycoproteins particularly, the NiV attachment glycoprotein (G) were the targets of predicted potential drug candidates for anti-viral attachment [73]. A therapeutic agent capable of preventing the attachment of these viruses to their respective host cell receptor would prevent the initiation of viral replication and early infection [74,75]. Viral surface proteins that bind to the surface receptor of the target cell are good candidates for vaccine development because they stimulate the production of blocking antibodies [76]. Various natural biomolecules are capable of blocking *in vitro* viral infection of a specific virus type, usually by blocking the replication machinery of the viral genome [43]. Whereas chloroquine is one of the very few potential agents that has been proposed as an option against preventing the host-cell attachment and entry of NiV and hMPV [77–79]. The present study assumed that blocking viral surface proteins can be achieved with natural biomolecules via AutoDock Vina, which utilises a global optimisation algorithm known as the Lamarckian Genetic Algorithm (LGA) to probe for the binding mode [80]. We selected four compounds of plant origin for which data in the literature have demonstrated antiviral properties against different viruses for the selected plant specimens [81–84]. Therefore, we tested their *in-silico* ability to interact with proteins involved in viral recognition of cell surface receptors of the main emerging viruses. Thus, we targeted the viral G envelope protein of NiV, in search of broad-spectrum antiviral molecules. Given its recent emergence in India and its neighbouring countries.

Prior studies have shown that extracts of *Cleistanthus* spp., *A. scorparia* and *T. orientalis* have potent antiviral activities against HIV-1, influenza A (H1N1 and H3N2 strains) and even SARS-CoV-2 by *Artemisia* spp. [34,37,81–83]. An investigation conducted by Ibrahim et al. revealed that cedrene and cedrol present in the essential oils of *Fortunella margarita* can inhibit the replication of the Influenza (H5N1) virus [84]. Moreover, β-cedrene present in the essential oils of *Artemisia argyi* was able to induce 50% inhibition of tobacco mosaic virus by preventing host cell attachment [85]. Another *in-silico* study indicated that cedrene can potentially inhibit SARS-CoV-2 [86]. Kodikara et al. was able to show that α-cedrene present in the essential oils of coriander can potentially inhibit SARS-CoV-2 [87]. Furthermore, another study showed that cedrol can induce antiviral activity against SARS-CoV-2 via the inhibition of viral replication [88]. Meanwhile, cedrol present in the essential oils of seven species of Lebanon plants was able to inhibit the replication of SARS-CoV-2 and HSV-1 [89]. Moreover, an *in vivo* study has revealed that digoxin being a steroid glycoside and similar digoxigenin can induce antiviral action against SARS-CoV-2 [90]. Studies have shown that components of plant based EOs can induce antiviral activities via the inhibition of viral hemagglutinin inhibitors, RNA polymerase, M2 ion channel proteins, protease, neuraminidase inhibitors, endosomal and lysosomal, non-structural proteins, caspase, glycoprotein/glycosylation and phospholipase inhibitors [91,92]. Moreover, few studies have shown the significance of the antiviral potential of plant based products against hMPV via *in vivo* and *in silico* [93–98]. Hypothetically, Figure 10 illustrates the proposed mode of antiviral action of digoxigenin, cedrene and cedrol, which promote blocking the viral surface proteins of NiV and hMPV, and inhibiting the entry and pathogenesis. Comparatively, ADME and toxicity studies of chloroquine, digoxigenin, cedrene and cedrol has been previously reported [99–103]. Studies have shown that EO compounds have the ability to bind with viral attachment and fusion glycoproteins and inhibit viral activity [104,105]. Meanwhile, digoxigenin has been used as a tool for probing viral glycoproteins [106]. MD simulations showed that when

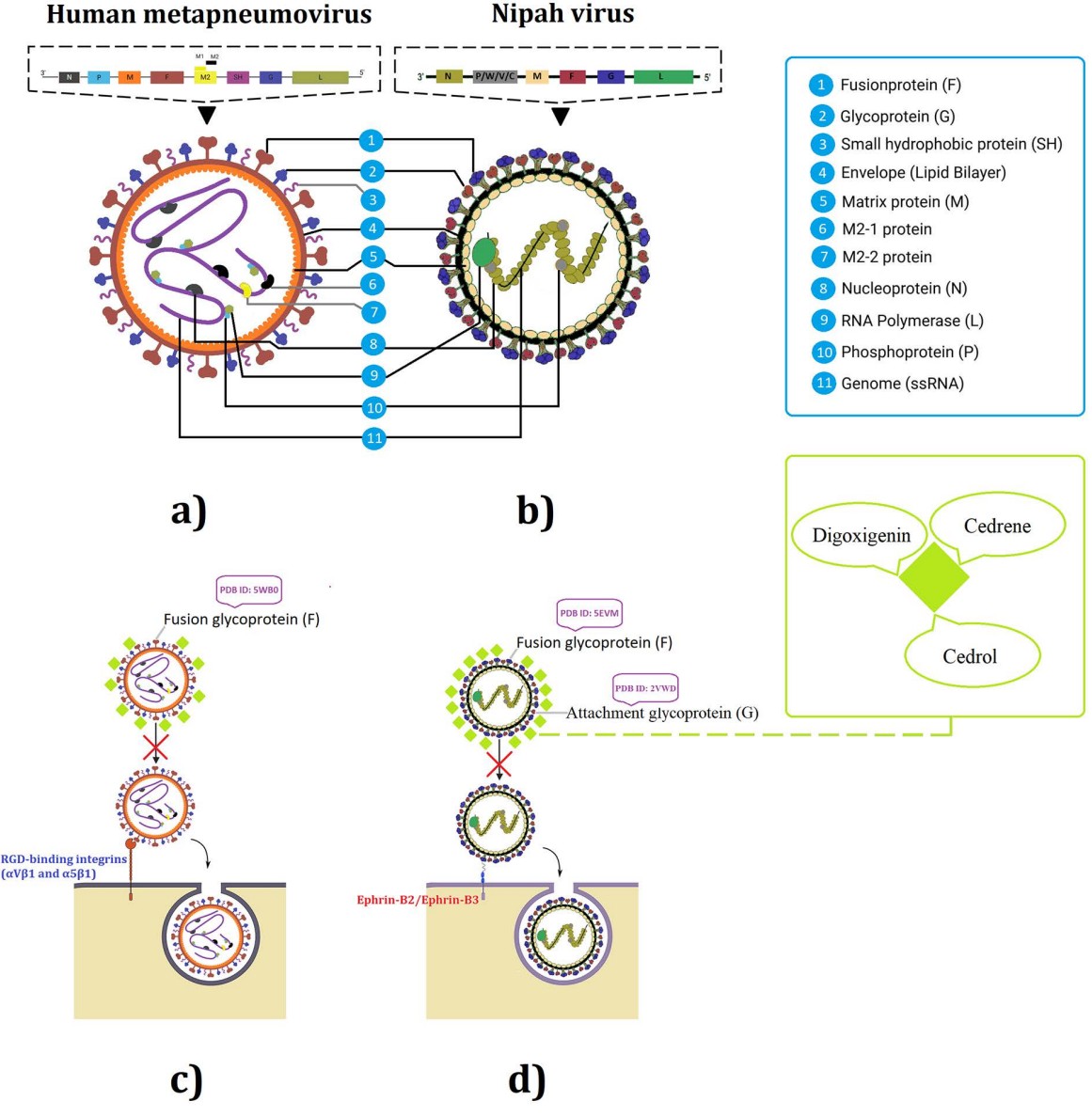

**Fig 10. Viron structure of hMPV (a) and NiV (b), and hypothetically proposed antiviral mechanism of action of digoxigenin, cedrene and cedrol with NiV attachment glycoprotein (G), fusion glycoprotein (F0) and hMPV fusion glycoprotein (F0), by targeting the interaction between viral surface proteins and RGD-binding integrins like αVβ1 and α5β1 receptors for hMPV (c) and cell receptors ephrin-B2/ephrin-B3 for NiV (d). These phytochemicals may contribute to hampering viral entry and mitigate pathogenesis in host cells.**

the drug receptor complex's structure changed over time, it stabilised in the physicochemical environment. The inhibitor stayed constant in spite of minor changes in loop and inner chain mobility. The selected ligand may be a promising lead chemical based on the structural stability of the docked complex after simulation tests. The protein-ligand complex achieved high stability across the 100 ns simulation time interval, which produced some intriguing results. Herein, simulating the real motions of atoms and molecules over time, MD simulations offer a thorough representation of the fluctuations and conformational changes observed in biological systems. *In silico* models are utilised in drug discovery

for purposes such as virtual screening to find promising drug candidates, forecasting physicochemical properties, and modelling disease processes, thereby saving time and cutting costs relative to conventional methods. Nevertheless, these models face limitations, including dependence on top-notch data (which might be lacking), possible inaccuracies in prediction algorithms and scoring functions, and the challenge of completely capturing the intricacies of biological systems, which keeps *in vitro* and *in vivo* validation crucial [107]. The use of *in silico* methods for predicting ADMET is crucial in the drug development process; however, these predictions face limitations, such as reliance on high-quality datasets, the intended application of the model, understanding of complex biological mechanisms, individual variability, and challenges in accurately forecasting toxicity. Moreover, early *in silico* tools encountered significant obstacles that limited their effectiveness, such as structure-based techniques like docking and molecular dynamics simulations, which struggled to make an impact in the ADMET domain due to the inherent promiscuity of many ADMET targets and the scarcity of high-resolution 3D structures. Likewise, pharmacophore models fell short in their prospective utility across the diverse landscape of chemical scaffolds, hampered by broad ligand specificity and the presence of multiple binding sites in a variety of ADMET targets. The overall effectiveness of these pioneering computational tools hinged on their ability to address the diverse needs encountered at different stages of drug discovery. Unfortunately, their predictive accuracy often proved inadequate for critical candidate selection. Moreover, progress in accurately predicting complex pharmacokinetic properties, such as clearance, volume of distribution, and half-life directly from molecular structure, was particularly sluggish, largely owing to a shortage of publicly available data. These limitations necessitate improvements in predictive methods and the integration of additional scientific approaches. [108–115]. Meanwhile, MM/PBSA data itself has entropy as the main contributor to Delta G, wherein entropy is mainly characterised by hydrophobic interactions, which makes the ligand cedrene as good as the others without requiring hydrogen bonds. These characteristics, which include conformational changes, stability, and binding affinity are essential for determining the complex system's biological importance. This information is very helpful in developing new medications and therapies as well as in deepening our understanding of biological molecules and their complexes.

## Conclusion

The present study aimed to find a prospective drug candidate against NiV and hMPV and compare the binding efficiency among the detected compounds via GC-MS by utilising compounds derived from *C. bracteosus* and essential oils of *A. scoparia* and *T. orientalis*. Hence, by the application of an *in silico* approach, we identified three phytochemical compounds from these plants with the predicted potential for anti-NiV and anti-hMPV activities. The lead compounds of these plants, digoxigenin, cedrol and cedrene exhibited remarkable binding affinities and strong interactions with key amino acid residues of NiV attachment glycoprotein (G), fusion glycoprotein (F0) of NiV and hMPV. The findings of this *in silico* study suggest that these compounds can be considered predicted potential antiviral agents to treat and eliminate NiV and hMPV by interfering with the viral host-cell recognition process. The outcome from GC-MS analysis combined with virtual molecular docking could assist researchers in designing or narrowing their search for bioactive phytoconstituents targeting the NiV and hMPV RBD. However, further experimental assessment and validation based on *in vitro*, *ex vivo* and *in vivo* studies are required to confirm the antiviral activity of the selected compounds against the NiV and hMPV RBD and to reveal more recommended characteristics related to these compounds and confirm the findings of our study, since these techniques remain predictive and cannot fully capture the complexity of biological environments, such as immune responses, enzymatic metabolism, and potential off-target effects of the selected ligands and their corresponding proteins. Furthermore, the lack of *in vitro* or *in vivo* experimental validation limits the confirmation of the actual efficacy, bioavailability, and safety of the proposed compounds. Therefore, additional experimental studies are necessary to validate these findings and facilitate progression toward clinical development, even though the computational data provide strong preliminary evidence and a valuable foundation.

## Supporting information

**S1 Table. Complete GC-MS profile of *C. bracteosus*.**
(DOCX)

**S2 Table. Complete GC-MS profiles of *A. scoparia* and *T. orientalis*.**
(DOCX)

**S1 Fig. Ramachandran plot showing residues at favoured region at 84.0% for 2VWD: NiV attachment glycoprotein (G) (a), 93.6% for 5EVM: NiV fusion glycoprotein (F0) (b), and 90.5% for 5WB0: hMPV fusion glycoprotein (F0) (c).**
(PDF)

## Acknowledgments

The authors in this paper are grateful to Dr. Fernando Berton Zanchi and his team affiliated to Laboratório de Bioinformática e Química Medicinal – LABIOQUIM of Brazil for conducting molecular dynamics simulations using high performance computing. Consent for publication: The study did not require any consent for publication.

## Author contributions

Software: Fernando Berton Zanchi.

Writing – original draft: Mackingsley Kushan Dassanayake.

Writing – review & editing: Teng-Jin Khoo, Chien Hwa Chong, Mohammed Tahir Ansari, Patrick Di Martino, Adam Figiel, Antoni Szumny, Omar Ashraf Elfar, Christophe Wiart, Rachael Symonds.

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
