## [Decision Letter · Decision Letter 0]

16 Jun 2025

Thank you for submitting your manuscript to PLOS ONE. After careful consideration, we feel that it has merit but does not fully meet PLOS ONE’s publication criteria as it currently stands. Therefore, we invite you to submit a revised version of the manuscript that addresses the points raised during the review process.

**ACADEMIC EDITOR:**

Dear Dr. Dassanayake,

I have received the recommendations regarding your manuscript. Based on the reviewers' report, I recommend that your manuscript undergo a thorough review. Please review each issue raised by the reviewers point by point and provide an updated version of your manuscript.

**Note from the Editorial Office:** One or more of the reviewers has recommended that you cite specific previously published works. Members of the editorial team have determined that the works referenced are not directly related to the submitted manuscript. As such, please note that it is not necessary or expected to cite the works requested by the reviewer.One or more of the reviewers has recommended that you cite specific previously published works. Members of the editorial team have determined that the works referenced are not directly related to the submitted manuscript. As such, please note that it is not necessary or expected to cite the works requested by the reviewer.One or more of the reviewers has recommended that you cite specific previously published works. Members of the editorial team have determined that the works referenced are not directly related to the submitted manuscript. As such, please note that it is not necessary or expected to cite the works requested by the reviewer.One or more of the reviewers has recommended that you cite specific previously published works. Members of the editorial team have determined that the works referenced are not directly related to the submitted manuscript. As such, please note that it is not necessary or expected to cite the works requested by the reviewer.

We look forward to receiving your revised manuscript.

Kind regards,

Mozaniel Santana de Oliveira, Ph.D

Academic Editor

PLOS ONE

Journal Requirements:

Reviewers' comments:

Reviewer's Responses to Questions

**Comments to the Author**

1. Is the manuscript technically sound, and do the data support the conclusions?

Reviewer #1: Partly

Reviewer #2: Yes

Reviewer #3: Partly

Reviewer #4: Yes

Reviewer #5: Yes

Reviewer #6: Yes

Reviewer #7: Yes

2. Has the statistical analysis been performed appropriately and rigorously?

Reviewer #1: No

Reviewer #2: No

Reviewer #3: N/A

Reviewer #4: Yes

Reviewer #5: No

Reviewer #6: Yes

Reviewer #7: Yes

3. Have the authors made all data underlying the findings in their manuscript fully available?

Reviewer #1: Yes

Reviewer #2: No

Reviewer #3: Yes

Reviewer #4: Yes

Reviewer #5: Yes

Reviewer #6: Yes

Reviewer #7: Yes

4. Is the manuscript presented in an intelligible fashion and written in standard English?

Reviewer #1: No

Reviewer #2: Yes

Reviewer #3: Yes

Reviewer #4: Yes

Reviewer #5: Yes

Reviewer #6: Yes

Reviewer #7: Yes

Reviewer #1: Good initiative. Requires thorough revision in a major way. Data presentation and analysis to be properly performed. More points are written in the document attached and is to be relayed by the editor.

Reviewer #2: This manuscript presents an in silico study investigating the potential antiviral activity of compounds from Cleistanthus bracteosus, Artemisia scoparia, and Thuja orientalis against Nipah virus (NiV) and human metapneumovirus (hMPV). The study utilizes GC-MS to identify major compounds, followed by molecular docking and molecular dynamics simulations to assess binding interactions with NiV and hMPV glycoproteins, and ADMET-AI analysis for pharmacokinetic and toxicity predictions. The findings suggest that digoxigenin, cedrene, and cedrol exhibit promising binding affinities to the viral targets.

While the study addresses an important public health issue (NiV and hMPV outbreaks) and explores the potential of natural products, which are relevant given the lack of approved treatments for NiV and limited treatment for hMPV, the manuscript requires significant revisions before it can be considered for publication. The primary concern is the overstatement of "anti-viral potential" based solely on in silico results without any experimental validation. The methodologies, particularly regarding ligand selection and MD simulation analysis, also need greater clarity. The abstract and conclusion should more accurately reflect the predictive nature and limitations of the in silico approach.

The title, abstract, introduction, results, and conclusion frequently use terms like "anti-Nipah virus and anti-human metapneumovirus potential" and suggest the compounds "have anti-Nipah virus and anti-human metapneumovirus potential" or "can be considered potential antiviral agents". These are strong claims based purely on computational predictions (binding affinity and predicted ADMET/toxicity profiles). In silico studies provide predictions and hypotheses, but they do not demonstrate actual antiviral activity. This requires experimental validation (e.g., in vitro cell culture assays). The language throughout the manuscript must be revised to accurately reflect that the study predicts or suggests potential activity based on computational modeling, emphasizing the need for future experimental validation. The title should be revised to reflect the predictive or in-silico nature of the findings more explicitly. Phrases like "predicted potential" or "in-silico analysis suggests potential" would be more appropriate.

Justify the rationale for selecting the four specific ligands (digoxigenin, lauric acid, cedrene, cedrol) is primarily based on their "high abundance" or being "major compounds" in the plant extracts, and "prominence of inducing a variety of broad-spectrum antimicrobial activities.

The GC-MS results list percentage abundances: cedrene (11.627%), cedrol (13.487%), digoxigenin (4.178%), and lauric acid (0.769%). While cedrene and cedrol are the most abundant in their respective essential oils, digoxigenin is only 4.178% and lauric acid a mere 0.769% in the C. bracteosus fraction. Justifying the selection of lauric acid based on "high abundance" appears inaccurate according to the provided data.

The connection between "broad-spectrum antimicrobial activities" and antiviral activity against specific viruses like NiV and hMPV needs to be clearly explained or justified. While some compounds may have both, the mechanisms are often different. Reference is made to previous studies on antimicrobial/antiviral properties, but the specific relevance of these particular compounds' previously known activities to the chosen viral targets and mechanisms (attachment and fusion glycoproteins) should be explored to strengthen the selection rationale.

Figure 1, mentioned as showing the GC-MS chromatograms, is not very clear and uninterpretable. Without the proper figure, it is difficult to fully appreciate the basis for identifying these specific compounds as "major" or understanding their relative abundance profiles. While the percentages are given in the text, enhancing the visualizing of the chromatograms would be helpful.

While binding affinities are provided for all tested ligands and targets, the interacting amino acid residues are only listed for digoxigenin, cedrene, and cedrol, not lauric acid or chloroquine. Including the interactions for lauric acid would complete the docking picture for all test ligands. Including those for the control (chloroquine) would allow for a comparison of how the experimental ligands interact with the targets relative to the reference drug.

Figures 2, 3, and 4 are described as showing 3D visualizations of molecular interactions, binding domains, binding sites, and interacting amino acids. These figures are crucial for visualizing the docking results and should be included. The text describes binding domains/sites (a, b) and interacting amino acids (c) for each complex, but the descriptions are general. More specific details on the nature of the interactions (hydrogen bonds, hydrophobic interactions, etc.) derived from the docking analysis would provide deeper insight into the binding mechanisms.

MD simulations were performed for compounds with binding affinities ≤ -6 kcal/mol. This threshold seems appropriate given the range of reported affinities.

The RMSD and RMSF analyses are presented with numerous specific chain identifiers (e.g., Chain E, Chain A, Chain B, Chain D, Chain F, Chain C). It is unclear what these chains represent within the context of the viral glycoproteins (e.g., are they different subunits, different monomers in a multimeric structure, or artifacts of the PDB file structure?). This needs clarification. Without this context, the specific peaks and average values reported for different chains are difficult to interpret.

The interpretation of the RMSD and RMSF results in the text appears somewhat contradictory. The abstract mentions "fluctuations at various points" but "remain mostly stable afterwards till 100 ns". The results section states that some chains showed "highest RMSD peak" at specific times but then discusses overall stability, like "remain around 0.15-0.3 nm without significant fluctuations till 100 ns" for 2VWD complexes. A more consistent interpretation of the simulation results, possibly referencing the specific regions of fluctuations (as implied by trajectory analysis), is needed. The correlation between specific structural changes (beta-turn-beta and helix-coil-helix alterations) mentioned in the abstract and the observed RMSD/RMSF fluctuations should be elaborated upon in the results and discussion.

Tables 3 and 4 present a wealth of ADMET and toxicity data. The discussion highlights several points, such as human tolerance, T. pyriformis toxicity, skin permeability, BBB/CNS permeability, Caco-2 permeability, P-glycoprotein, CYP enzymes, AMES toxicity, and hERG inhibition, often comparing test ligands to chloroquine.While a general comparison is made, a more in-depth interpretation of what these properties mean for potential drug development is necessary. For example, "higher skin permeability than -2.5 log Kp" is stated, but the significance of this value and whether it is considered high or low relative to desirable drug properties should be clarified. Similarly, the significance of T. pyriformis toxicity (> -0.5 log ug/L) and Minnow toxicity (> 0.246 log mM) in the context of human drug safety needs explanation.

The statement that digoxigenin, cedrene, and cedrol "showed outstanding characteristics that were potentially beneficial during a clinical trial, since they did not indicate AMES toxicity" is a strong claim. While avoiding AMES toxicity and hERG inhibition are desirable properties, predicting "beneficial characteristics during a clinical trial" solely from these in silico predictions is speculative and should be revised.

Minor comments

The introduction provides background on NiV and hMPV. The discussion of hMPV classification (A1, A2a, A2b, B1, B2, with gene duplication variants A2b1, A2b2, A2c) and the need for nomenclature reassessment is detailed but the relevance of this genotypic diversity to the chosen hMPV target protein (fusion glycoprotein F0, PDB ID 5WB0) is not explicitly stated. Is PDB ID 5WB0 representative of a specific hMPV subtype, and does this classification impact the generalizability of the findings across different hMPV strains?

The discussion could benefit from a more critical evaluation of the in silico methods used and their limitations. While it mentions that in silico methods are low-cost and safe for screening, it should also acknowledge that these are predictive tools and the results are theoretical until experimentally validated. The comparison of findings with previous studies is good, but linking the previously reported activities of these compounds (or related ones) to the current study's in silico findings against NiV/hMPV targets could be more direct and analytical.

There appear to be some formatting inconsistencies and potential typos (e.g., line breaks mid-sentence in some sections, "weas" instead of "was" in RMSF description). These should be corrected during revision.

Ensure all statements drawn from sources are cited. For example, the statement in the abstract "The three out of four compounds tested exhibited remarkable binding affinities..." and the specific affinity values should be cited.

Reviewer #3: The manuscript lacks the scientific strength or depth and rigor, and is not up to the standards for publishing in PLOS One. However, the manuscript can be improved and following points need to be addressed -

1. INTRODUCTION

I. In introduction line 111 and 132-133 are basically repetition. Please avoid repeating the same info. Revise line 160.

II. A genomic organization and structure of the virus is highly recommended.

III. While literature about the viral background is detailed, the link to drug discovery is somewhat lacking.

IV. Justification of selecting these 3 plants are not well contextualized. Why do the authors think that these plants would give antiviral activity or have potential antiviral activity particularly against these 2 viruses? These have to be mentioned in the introduction with suitable references.

V. The discussion on the evolution of hMPV should be very brief. It tends to distract the focus of the study objective. Rather more information on the traditional, pharmacological and phytochemicals of the plant would have been better match with the scope of this study.

VI. I am not sure whether the term ‘broad spectrum’ is appropriate to use here. Why the authors would claim as broad spectrum?

VII. The objectives could be more clear and specific.

2. Methodology

I. How the bioactivity of the TLC fraction were obtained? The solvents used for extraction and fractionation are both toxic and how this fractions are clinically significant in practical application?

II. What drove the authors to analyse the phytochemicals by GC-MS since it will provide the information about non-polar, volatile compounds. Is there any evidence or reports that directed to focus on these types of compounds?

III. The docking experments should be validated. No reference compound/drug was included to compare with. This is important to understand the possible anti viral activity of these phytocompounds.

IV. Did the author included the co-crystalized ligand in the docking? It would help in validating the docking work. Moreover, the compounds bound to the amino acid residues at the binding pocket but it is not discussed how significant are these residues. Are these residues important for the activity?

3. Results

I. The quality of the figures is not good. Should provide good resolution images.

II. For the MD simulation, it is important to add calculations for more parameters like, MMBSA/MMPBSA, radius of gyration, hydrogen bonding, PCA analysis to understand the binding energy convergence, or conformational dynamics of the ligand-protein complexes.

III. Why the authors did not provide RMDS and RMSF data for the apo protein? Inclusion of reference drug is also important to compare the binding stability for these compounds and understand the potential activity.

Reviewer #4: The data support the conclusion. it is a well-written paper, perhaps a bit too long in the introduction, but very explanatory. The data presented clear information, and the statistical analysis was also well executed.

Reviewer #5: This manuscript presents an in silico study using AI-driven molecular docking, molecular dynamics simulations, and ADMET predictions to evaluate the antiviral potential of major phytochemicals from Cleistanthus bracteosus, and essential oils from Artemisia scoparia and Thuja orientalis against Nipah virus (NiV) and human metapneumovirus (hMPV). The study identifies several compounds with higher predicted binding affinities than chloroquine, a known antiviral agent, and discusses their interaction profiles and predicted pharmacokinetic properties. However, I have some reservations listed below:

1. The study is entirely in silico. While computational predictions are valuable for hypothesis generation, the absence of any in vitro or in vivo validation limits the immediate impact and practical applicability of the findings.

2. Only a small number of compounds (four) were selected for detailed docking and simulation, which may not fully represent the antiviral potential of the source plants and oils.

3. The manuscript references "AI-driven" methods (e.g., ADMET-AI), but the specific AI models, their validation, and their performance metrics are not described in detail, which may limit reproducibility and assessment of the robustness of the predictions.

4. The abstract and conclusion suggest these compounds could be used as novel therapeutic strategies, but without experimental validation, such statements should be more cautiously framed.

5. There are minor grammatical errors and awkward phrasings throughout the abstract and initial sections, which may hinder readability and should be addressed in revision.

6. Chloroquine is used as a control, but its relevance as a comparator for NiV and hMPV (for which it is not a standard therapy) should be justified or alternative controls considered.

Reviewer #6: Comments for PONE-D-25-23763

I have gone through the assigned manuscript entitled “Major secondary metabolites present in Cleistanthus bracteosus Jabl. and essential oils of Artemisia scoparia and Thuja orientalis have anti-Nipah virus and anti-human metapneumovirus potential: An AI-driven in-silico study”, and found that it is competently well-written and the manuscript contains sufficient data, but it should meet the following comments:

Major comments-

1- Title should be shortened.

2. The abstract should provide a concise summary of the key findings of the whole study.

3- Authors added all old references in the Introduction, so I suggest adding more recently published work with your relevant studies (2023-2025) and rewriting the Introduction section.

4- The material and methods should include more details.

5- The authors have to compare their results with literature data and improve the results and the discussion section completely.

6- Follow the unit in the same system throughout the manuscript.

7- Why did the author use both NiV and (hMPV proteins for docking studies? Give your appropriate reasons properly and separately. Your explanation should be added to the manuscript's results and discussion sections.

8- Mention which docking algorithm and scoring function were utilized in the docking process.

9- How do authors validate the protein? Add a Ramachandran plot to illustrate the distribution of phi (ϕ) and psi (ψ) dihedral angles for the protein's residues.

10- In the results and discussion, the authors should strengthen the explanation of the validity of the native ligand docking results.

11- Authors are suggested to add the following articles to the manuscript.

**Deleted by editor.**

12- References are not properly cited and should follow the journal's style. The References section has to be revised after all corrections.

13-In the manuscript, there are many spelling and grammatical errors. So, grammatical and punctuation errors must be corrected. I suggest improving the English language.

14-Conclusion should be succinct and precise, not be same as the Abstract.

My Opinion: Major Revision.

Reviewer #7: Reviewer’s Comment

Manuscript Number: PONE-D-25-23763

Title: Major secondary metabolites present in Cleistanthus bracteosus Jabl. and essential oils of Artemisia scoparia and Thuja orientalis have anti-Nipah virus and anti-human metapneumovirus potential: An AI-driven in-silico study

Review Summary

This study deals with the investigation of anti-Nipah virus and anti-human metapneumovirus potential: of major secondary metabolites present in Cleistanthus bracteosus Jabl. and essential oils of Artemisia scoparia and Thuja orientalis. This present study offers computational insights into the nti-Nipah virus and anti-human metapneumovirus potential of these metabolites by using an AI-driven in silico approach. The binding affinities further revealed that some of these compounds interacted more strongly with NiV and hMPV receptors than the drug control chloroquine (-5.5 to -6.1 kcal/mol). Moreover, MD simulations illustrated phytochemical interacting amino acid residues associated with each receptor of NiV and hMPV. These phytochemical compounds were further evaluated using ADMET-AI platforms. The research is relevant, however, several aspects of the study, particularly in terms of methodological clarity and scientific rigor, require significant improvement for the manuscript to meet publication standards.

• The keywords are too long and too many. It is recommended to limit the keywords between four to six

• Line 70, there should be a comma after the word “particular”

• Below line 186 under the materials and method, authors should comment on the grade of reagents and solvents used or report on the purity level of these reagents and solvents

• Line 215, kindly justify why extraction temperature as high as 100oC was used knowing fully well that most of the bioactive metabolites are heat-labile

• Line 219, the description of the rotary evaporation should the phrase “under reduced pressure”

• Line 494: put the expression “in vitro” in italics

• Line 543, the word activity should be in plural form to become “anti-NiV and anti-hMPV activities”

• Line544, there should be a comma after the word “particularly”

• Line 778 Ref 67: put the expression “in vivo” in italics

• Line 789 Ref 71: put the expression “in vitro” in italics

• Line 826 Ref 84: put the expression “in vitro” in italics

• Line 606 Ref 2, the article title should be in lower case to become “Pandemics throughout history”

• Line 617 Ref 8, the article title should be in lower case to become “Virus transmission from bats to humans associated with drinking traditional liquor made from Date Palm Sap, Bangladesh”

• Line 675 Ref 28, kindly put the name “Cleistanthus collinus” in italics being a botanical name.

• Line 677 Ref 29: kindly put the name “Artemisia scoparia” in italics being a botanical name.

• Line 683 Ref 32: the article title should be in lower case to become “Phytochemical 684 profiling, antioxidant, antimicrobial and cholinesterase inhibitory effects of essential 685 oils isolated from the leaves of Artemisia scoparia and Artemisia absinthium

• Line 685 Ref 32: kindly put the names “Artemisia scoparia” and “Artemisia absinthium” in italics being botanical names.

.

Reviewer #1: No

Reviewer #2: **Yes:**Shafi Ullah KhanShafi Ullah KhanShafi Ullah KhanShafi Ullah Khan

Reviewer #3: **Yes:**AKM Moyeenul HuqAKM Moyeenul HuqAKM Moyeenul HuqAKM Moyeenul Huq

Reviewer #4: No

Reviewer #5: No

Reviewer #6: No

Reviewer #7: **Yes:**Olayinka Oyewale AJANIOlayinka Oyewale AJANIOlayinka Oyewale AJANIOlayinka Oyewale AJANI

---

## [Author Response · Author response to Decision Letter 1]

4 Dec 2025

A separate file has been attached for response to reviewers. All reviewers comments has been addressed.

---

## [Decision Letter · Decision Letter 1]

2 Jan 2026

Dear Dr.  Dassanayake,

Thank you for submitting your manuscript to PLOS ONE. After careful consideration, we feel that it has merit but does not fully meet PLOS ONE’s publication criteria as it currently stands. Therefore, we invite you to submit a revised version of the manuscript that addresses the points raised during the review process.

**ACADEMIC EDITOR: Dear authors, I have received the required number of revisions to your manuscript and have made a decision. Please review each point raised by the reviewers.**  **Best regards****Mozaniel de Oliveira Ph.D.****Academic Editor.**==============================

We look forward to receiving your revised manuscript.

Kind regards,

Mozaniel Santana de Oliveira, Ph.D

Academic Editor

PLOS One

Journal Requirements:

Reviewer's Responses to Questions

**Comments to the Author**

Reviewer #2: All comments have been addressed

Reviewer #5: All comments have been addressed

Reviewer #7: All comments have been addressed

2. Is the manuscript technically sound, and do the data support the conclusions?

Reviewer #2: Yes

Reviewer #5: Partly

Reviewer #7: Yes

3. Has the statistical analysis been performed appropriately and rigorously?

Reviewer #2: No

Reviewer #5: No

Reviewer #7: Yes

4. Have the authors made all data underlying the findings in their manuscript fully available?

Reviewer #2: No

Reviewer #5: Yes

Reviewer #7: Yes

5. Is the manuscript presented in an intelligible fashion and written in standard English?

Reviewer #2: Yes

Reviewer #5: Yes

Reviewer #7: Yes

Reviewer #2: RMSD fluctuations are due to structural changes in the beta-turn-beta and helix-coil-helix alterations. Although noted as a "generalized statement", providing explicit evidence (even brief qualitative examples from the trajectory analysis) that these specific structural regions drive the observed RMSD peaks would strengthen the analysis, rather than relying solely on the general reference.

Though author clarified that lauric acid (0.769%) was selected because it was present in the most bioactive TLC isolated fraction, the introduction or results section should explicitly reinforce that its selection overrides the low abundance percentage due to its detected presence in this highly active fraction, rather than being generally considered "high abundance".

Just like before concern as multiple reviewers noted that the most of the figures (including the GC-MS chromatograms and MD plots) were difficult to read. Before final acceptance confirmation is needed that the high-resolution PNG/vector graphic files submitted separately to render all details

Reviewer #5: The authors have addressed the reviewer comments but some needs to be corrected.

1.The manuscript remains longer than necessary, and several results—particularly MD analyses—are repetitive and should be condensed.

2.Molecular dynamics results should be presented more integratively, focusing on overall stability trends rather than detailed metric-by-metric narration.

3.Claims implying therapeutic efficacy should be moderated to reflect the predictive, in-silico nature of the study.

4.The ADMET discussion should more clearly acknowledge the limitations and preliminary nature of computational pharmacokinetic predictions.

5.The Abstract would benefit from reduction in length and numerical detail for improved clarity.

6.Some sections still contain long, complex sentences; a final language polish is recommended.

7.Conceptual mechanism figures should be explicitly labeled as hypothetical.

8.Read and cite below papers:

Patel, C. N., & Mall, R. (2025). Repurposing drugs for monkeypox using artificial intelligence. The Scientific Basis of Mpox (Monkeypox), 421-440.

Patel, C. N., Jani, S. P., Kumar, S. P., Modi, K. M., & Kumar, Y. (2022). Computational investigation of natural compounds as potential main protease (Mpro) inhibitors for SARS-CoV-2 virus. Computers in Biology and Medicine, 151, 106318.

Patel, C. N., Mall, R., & Bensmail, H. (2023). AI-driven drug repurposing and binding pose meta dynamics identifies novel targets for monkeypox virus. Journal of Infection and Public Health, 16(5), 799-807.

Reviewer #7: Reviewer’s Comment

Manuscript Number: PONE-D-25-023763R1

Title: Predicted antiviral potential of phytochemicals prolific in Cleistanthus bracteosus Jabl.

and essential oils of Artemisia scoparia and Thuja orientalis against Nipah virus and

Human metapneumovirus: An AI-driven in-silico study

Review Summary

This study deals with the utilization of major compounds present in Cleistanthus bracteosus and in the essential oils extracted from Artemisia scoparia and Thuja orientalis to analyse their antiviral activity against the NiV and hMPV through computational approach. Molecular docking and molecular dynamics simulations were the primary tools used by these authors to assess the binding interactions of the major compounds detected by GC-MS analysis. These phytochemical compounds were further evaluated using ADMET platforms. The authors also reported that RMSD graphs show fluctuations at various points in the protein-ligand complex system due to structural changes in the beta-turn-beta and helix-coil-helix alterations at specific points resulting in increased RMSD within a time frame of 50 ns. However, these alterations in the simulation remain mostly stable afterwards till 100 ns.

Strengths of the Manuscript

1. Comprehensive Computational Workflow:

The study employs a multi-tiered in silico strategy; molecular docking, molecular dynamics simulations, binding free energy calculations (MM/PBSA), and ADMET profiling, which provides a robust and layered assessment of the compounds’ potential. This methodological rigor is consistent with current standards in computational drug discovery.

2. Relevant and Timely Virological Targets:

Nipah virus (NiV) and human metapneumovirus (hMPV) are both significant public health threats with limited therapeutic options. Targeting their entry glycoproteins (G and F) is a rational and promising antiviral strategy, well justified in the introduction.

3. Use of Control Compound:

Inclusion of chloroquine as a reference adds context to the docking scores and ADMET results, allowing readers to gauge the relative promise of the phytochemicals.

4. Integration of AI and Machine Learning Tools:

The use of AI-driven ADMET platforms (e.g., pkCSM, SwissADME) and modern simulation software (GROMACS, AMBER) reflects contemporary, state-of-the-art practices in computational pharmacology.

5. Detailed Supporting Analyses:

The manuscript includes RMSD, RMSF, SASA, Rg, and hydrogen bond analyses over 100 ns MD simulations, which lends credibility to the stability and interaction patterns of the protein–ligand complexes.

6. Clear Presentation of Data:

Results are well-organized in tables and figures, and the discussion contextualizes findings within existing literature on antiviral phytochemicals.

7. Ethnobotanical Relevance:

The selection of Cleistanthus bracteosus, Artemisia scoparia, and Thuja orientalis is supported by traditional use and prior evidence of bioactive properties, adding translational relevance.

Originality and Novelty

1. First Reported Investigation Against NiV and hMPV:

To the best of current knowledge, this is the first in silico study to specifically evaluate digoxigenin, cedrene, and cedrol, which is done individually or in combination, against the attachment and fusion glycoproteins of both NiV and hMPV. This represents a novel contribution to the field of antiviral phytochemistry.

2. Cross-Kingdom Phytochemical Exploration:

The combination of compounds from three distinct plant sources which are, a flowering plant, a conifer, and a medicinal herb, broadens the chemical diversity examined and highlights the potential of integrative phytochemical sourcing.

3. AI-Enhanced Predictive Workflow:

While molecular docking and MD are common, the explicit incorporation of AI-based ADMET prediction early in the screening pipeline is a forward-looking approach that aligns with modern drug discovery trends.

4. Hypothesis-Driven Mechanistic Insight:

The proposed mechanism (blocking viral entry via glycoprotein interference) is clearly illustrated and supported by docking and dynamics data, offering a testable model for future experimental validation.

Overall Assessment of the Manuscript

This manuscript presents a well-structured, technically sound, and novel computational study that meets the standards for publication in a reputable journal. Its strengths lie in:

• The use of a multi-method computational pipeline that validates findings across several layers of analysis.

• The focus on understudied viral targets with high clinical relevance.

• The integration of traditional botanical knowledge with modern AI-driven tools.

• The clear, hypothesis-driven approach that can guide future in vitro and in vivo studies.

Suggested Few Amendments for Enhancement:

• Clarify the % abundance of digoxigenin in C. bracteosus (4.178% is noted but considered “major”, brief justification would help).

• Ensure all figure and table references in the text are correctly placed in the final version.

• Consider briefly discussing the limitations of in silico predictions in the conclusion to frame the findings appropriately.

Recommendation: Accept

The manuscript demonstrates significant originality in its target–compound combinations and employs a rigorous, contemporary computational methodology. It provides compelling preliminary evidence that digoxigenin, cedrene, and cedrol warrant further experimental investigation as potential broad-spectrum antiviral agents. The work is suitable for publication in a peer-reviewed journal, particularly those focusing on virology, computational biology, or natural product drug discovery. The manuscript has the potential to make a valuable contribution to the field. Hence, from my own opinion, the manuscript could be accepted for publication in Plos ONE.

.

Reviewer #2: No

Reviewer #5: No

Reviewer #7: **Yes:**Olayinka Oyewale AJANIOlayinka Oyewale AJANIOlayinka Oyewale AJANIOlayinka Oyewale AJANI

---

## [Decision Letter · Decision Letter 2]

25 Feb 2026

Dear Dr. Dassanayake,

Thank you for submitting your manuscript to PLOS ONE. After careful consideration, we feel that it has merit but does not fully meet PLOS ONE’s publication criteria as it currently stands. Therefore, we invite you to submit a revised version of the manuscript that addresses the points raised during the review process.

**ACADEMIC EDITOR:** 

Dear Dr. Dassanayake,

Thank you for sending the revised version of your manuscript. After the peer review process, some additional important revisions were requested.

Please carefully consider the reviewer’s comments and provide an updated version of your manuscript addressing all the points raised.

We look forward to receiving your revised manuscript.

Kind regards,

Mozaniel Santana de Oliveira, Ph.D

Academic Editor

PLOS One

Journal Requirements:

Reviewers' comments:

Reviewer's Responses to Questions

**Comments to the Author**

Reviewer #5: All comments have been addressed

Reviewer #7: All comments have been addressed

2. Is the manuscript technically sound, and do the data support the conclusions?

Reviewer #5: Yes

Reviewer #7: Yes

3. Has the statistical analysis been performed appropriately and rigorously?

Reviewer #5: Yes

Reviewer #7: Yes

4. Have the authors made all data underlying the findings in their manuscript fully available?

Reviewer #5: Yes

Reviewer #7: Yes

5. Is the manuscript presented in an intelligible fashion and written in standard English?

Reviewer #5: Yes

Reviewer #7: Yes

Reviewer #5: The authors have substantially addressed the reviewer comments, but the revision is not yet final-submission ready. The remaining issues are presentation, tone, and framing, not missing analyses or methodology.

Reviewer #7: Revised Reviewer’s Comment

Manuscript Number: PONE-D-25-023763R2

Title: Predicted antiviral potential of phytochemicals prolific in Cleistanthus bracteosus Jabl. and essential oils of Artemisia scoparia and Thuja orientalis against Nipah virus and Human metapneumovirus: An AI-driven in-silico study

This study employs a comprehensive computational approach to evaluate the antiviral potential of major phytochemicals from Cleistanthus bracteosus, Artemisia scoparia, and Thuja orientalis against Nipah virus (NiV) and human metapneumovirus (hMPV). Utilizing molecular docking, molecular dynamics (MD) simulations, binding free energy calculations (MM/PBSA), and AI-driven ADMET profiling, the authors provide a robust, multi-layered analysis. The manuscript is well-structured, methodologically sound, and addresses a significant gap in the search for therapeutics against these high-priority viral pathogens.

Strengths of the Manuscript

1. Comprehensive Computational Workflow: The multi-tiered in silico strategy provides a rigorous assessment, aligning with current standards in computational drug discovery.

2. Relevant and Timely Virological Targets: The focus on the entry glycoproteins of NiV and hMPV, both significant public health threats with limited treatments, is highly rational and justified.

3. Use of a Control Compound: The inclusion of chloroquine as a reference provides valuable context for interpreting docking scores and ADMET results.

4. Integration of Modern Tools: The use of AI-driven ADMET platforms and advanced simulation software reflects state-of-the-art practices.

5. Detailed Supporting Analyses: Extensive MD simulation analyses (RMSD, RMSF, SASA, Rg, hydrogen bonds) over 100 ns credibly support the stability and interaction patterns of the complexes.

6. Clear Presentation of Data: Results are well-organized and effectively supported by tables and figures, with a discussion that appropriately contextualizes the findings.

7. Ethnobotanical Relevance: The plant selection is supported by traditional use and prior evidence of bioactivity, adding translational value.

Originality and Novelty

1. First Reported Investigation: To current knowledge, this is the first in silico study to specifically evaluate digoxigenin, cedrene, and cedrol against the attachment and fusion glycoproteins of both NiV and hMPV.

2. Cross-Kingdom Phytochemical Exploration: Examining compounds from three distinct plant sources broadens the chemical diversity investigated.

3. AI-Enhanced Predictive Workflow: The explicit incorporation of AI-based ADMET prediction strengthens the screening pipeline.

4. Hypothesis-Driven Mechanistic Insight: The proposed mechanism of blocking viral entry is clearly illustrated and well-supported by the computational data.

Author Revisions and Final Assessment

The authors have thoughtfully addressed all minor amendments suggested in the previous review:

• They have clarified the justification for classifying digoxigenin (4.178%) as a major compound.

• All figure and table references in the text have been verified and correctly placed.

• A brief discussion on the inherent limitations of in silico predictions has been added to the conclusion, appropriately framing the study's findings.

Overall Recommendation

All necessary corrections have been satisfactorily carried out. The manuscript presents a novel, technically robust, and well-executed computational study that makes a valuable contribution to the field. Its strengths in methodology, relevance, and clarity significantly outweigh any inherent limitations of the in silico approach. The work is now suitable for publication and is recommended for acceptance in its current form.

.

Reviewer #5: No

Reviewer #7: **Yes:**Olayinka Oyewale AJANIOlayinka Oyewale AJANIOlayinka Oyewale AJANIOlayinka Oyewale AJANI

---

## [Author Response · Author response to Decision Letter 3]

25 Feb 2026

Responses to Reviewers has been attached in a separate file.

Response from author: Made corrections to few typos and sentences from lines 265-268. Apart from that, it is difficult to determine the flaws in presentation, tone, and framing as mentioned by Reviewer #5. Since Reviewer #7 and other reviewers have accepted the manuscript in its current form, I would very much appreciate if Reviewer #5 can specify the corrections that needs to be for presentation, tone, and framing?

---

## [Editor Report · Decision Letter 3]

17 Mar 2026

Predicted antiviral potential of phytochemicals prolific in Cleistanthus bracteosus Jabl. and essential oils of Artemisia scoparia and Thuja orientalis against Nipah virus and Human metapneumovirus: An AI-driven in-silico study

PONE-D-25-23763R3

Dear Dr. Dassanayake,

We’re pleased to inform you that your manuscript has been judged scientifically suitable for publication and will be formally accepted for publication once it meets all outstanding technical requirements.

Kind regards,

Mozaniel Santana de Oliveira, Ph.D

Academic Editor

PLOS One
---

## [Editor Report · Acceptance letter]

PONE-D-25-23763R3

PLOS One

Dear Dr. Dassanayake,

I'm pleased to inform you that your manuscript has been deemed suitable for publication in PLOS One. Congratulations! Your manuscript is now being handed over to our production team.

Kind regards,

on behalf of

Dr. Mozaniel Santana de Oliveira

Academic Editor

PLOS One